# DiffNCL: Diffusion-Driven Weakly-Noisy Correspondence Learning

## Abstract

Current noisy correspondence learning (NCL) pipelines typically treat correspondence quality as a binary variable, neglecting the abundant category of *weakly-noisy correspondences*. Two persistent issues are introduced: (i) *over-exclusion*, where partially informative pairs are discarded as negatives, shrinking the effective data manifold, and (ii) *under-alignment*, where residual noise from weakly mismatched pairs propagates through gradient updates, degrading representation fidelity. To address these challenges, this work pioneers a unified forward–reverse diffusion framework called "**DiffNCL**" to explicitly amplify and subsequently purify weakly noisy correspondences for robust noisy correspondence learning. In the forward diffusion, synchronized stochastic perturbations inject Gaussian noise into paired visual–textual embeddings, and step-wise similarities are aggregated to highlight the diffusion discrepancy of weakly noisy mismatches. During reverse diffusion, two complementary consistency objectives, i.e., intra-modal structural consistency and cross-modal semantic consistency, progressively purify and reconstruct weakly noisy correspondences into high-quality pairs for subsequent training cycles. Extensive experiments on benchmark datasets, including Flickr30K, MS-COCO, and CC152K, are conducted to demonstrate the superiority of DiffNCL over state-of-the-art baselines for cross-modal retrieval against noisy correspondences.

## 1 Introduction

With the exponential growth of multimedia data, cross-modal retrieval (Diao et al., 2021; Cheng et al., 2022; Fu et al., 2023; Pham et al., 2024; Lin et al., 2024) has emerged as a critical research focus in both academic and industrial communities. Despite demonstrating significant success across multiple domains, existing cross-modal approaches face challenges due to real-world datasets frequently containing noisy correspondences (Huang et al., 2021) arising from non-specialist annotations or collection from unreliable web sources in practical implementations (Sharma et al., 2018; Jia et al., 2021). Noisy correspondence, defined as persistent misalignment between semantically paired modalities, has severely compromised the effectiveness of conventional cross-modal methods that rely on perfectly aligned image-text pairs (Han et al., 2023; Yang et al., 2023; Qin et al., 2023), ultimately limiting their real-world applicability.

Noisy correspondences corrupt contrastive training by injecting false positives and skewing gradient directions, leading to distorted embeddings and degraded retrieval performance. Conventional noisy correspondence learning (NCL) remedies (Huang et al., 2021; Qin et al., 2022; Han et al., 2023; Yang et al., 2023; Ma et al., 2024), e.g., manual data curation (Sharma et al., 2018), strict negative sampling (Yang et al., 2023), and robust loss functions (Han et al., 2023), effectively remove extreme misalignments but often over-exclude informative pairs. Objective reweighting (Huang et al., 2021) and curriculum learning (Qin et al., 2023) offer coarser mitigation by down-weighting or iteratively filtering noisy samples, yet they still operate on a binary clean-vs-noisy basis. In recent years, some advanced works (Dang et al., 2024; Duan et al., 2024; Feng et al., 2023; Han et al., 2024) exploit the memorization effect of deep neural networks, where simple patterns are learned before fitting noise, to distinguish clean samples from noisy ones. Despite recent advances, a binary clean-vs-noisy paradigm fails to capture weakly-noisy correspondences—partially aligned pairs that, despite minor mismatches, carry valuable semantic information. As shown in Figure 4, weakly-noisy correspondences occupy the gray area between perfectly matched and fully corrupted pairs. Discarding them wastes rich cross-modal cues, while treating them as clean introduces subtle noise.

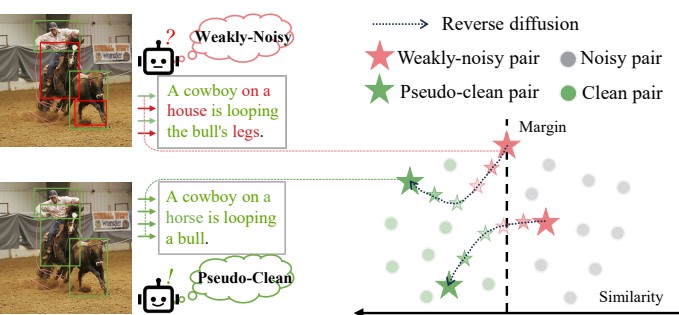

Figure 1: Illustration of weakly-noisy correspondences converted into pseudo-clean by DiffNCL. Weakly-noisy correspondences are partially aligned cross-modal data that lie between perfectly matched (clean) and fully corrupted (noisy) pairs, with minor semantic mismatches but valuable semantic cues. DiffNCL aims to turn these weakly-noisy pairs into high-fidelity pseudo-clean representations to address over-exclusion and under-alignment issues.

By treating weakly-noisy correspondences as either fully clean or entirely noisy, existing solutions still suffer two intertwined failures: (i) **over-exclusion** erases valuable cross-modal cues, narrowing the data manifold and hampering generalization, while (ii)**under-alignment** allows misalignments to contaminate parameter updates, slowing convergence and degrading embedding quality.

To address these challenges, we propose a novel **Diff**usion-Driven Weakly-**N**oisy **C**orrespondence **L**earning (**DiffNCL**) framework, that harnesses a forward–reverse diffusion process, i.e., forward diffusion for discrepancy mining and weakly-noisy pair identification, and reverse diffusion with consistency constraints for denoising and pseudo-clean representation generation, to robustly mitigate noisy cross-modal correspondences. In the **forward diffusion** stage, synchronized Gaussian noise is injected into visual and textual features following a pre-defined schedule, ensuring the similarities of cross-modal features reflect distributional differences among clean, weakly-noisy, and noisy instances in the diffusion flow. For each diffusion step, cosine similarities are computed and aggregated to derive stability-weighted diffusion discrepancies, enhancing discrimination of weakly-noisy samples. In the **reverse diffusion** phase, modality-specific denoisers transform noisy features into pseudo-clean representations under two consistency objectives, i.e., Intra-modal structural consistency and Cross-modal semantic consistency. On the one hand, the proposed intra-modal structure consistency preserves the intrinsic discriminative topology of denoised features and maintains semantic stability before and after denoising, thus preventing semantic collapse. On the other hand, cross-modal semantic consistency drives denoised features toward the clean manifold while penalizing high similarity with unrelated original features, thereby inhibiting the propagation of weakly-noisy correspondences. Through end-to-end training in an end-to-end manner, the reverse diffusion stage maps corrupted inputs into high-fidelity pseudo-clean representations. By substituting raw noisy features with these pseudo-clean embeddings in the retrieval objective, DiffNCL achieves robust training that effectively mitigates weakly-noisy correspondences. The main contributions are summarized as follows:

- Our work pioneers the integration of diffusion dynamics into noisy correspondence learning by proposing DiffNCL. To the best of our knowledge, this is *the first attempt* to tackle cross-modal noisy correspondence learning with a unified forward–reverse diffusion process.

- We design a forward diffusion–based data partitioning strategy that derives diffusion discrepancies by dynamically analyzing feature similarity gradients during a predefined diffusion schedule and applying stability-weighted fusion to capture evolving visual–textual semantic distributions, thereby improving data partitioning accuracy in noisy environments.

- We propose a reverse diffusion–based denoising reconstruction paradigm that leverages dual diffusion consistency constraints, i.e., intra-modal structural and cross-modal semantic consistency, to iteratively convert weakly-noisy features into high-fidelity pseudo-clean representations, enhancing the robustness of cross-modal correspondence training.

- Extensive experiments on synthetically and real-world noisy image-text benchmark datasets demonstrate that DiffNCL outperforms existing robust methods in handling weakly-noisy correspondences, verifying its effectiveness in suppressing noise interference.

## 2 RELATED WORKS

### 2.1 CROSS-MODAL RETRIEVAL

As a fundamental task in multimedia research, cross-modal retrieval aims to query for the relevant items across different modalities. Existing cross-modal retrieval methods can be broadly categorized into two main approaches: 1) Coarse-grained approaches (Fu et al., 2023; Li et al., 2022; Chen et al., 2021; Li et al., 2019; Faghri et al., 2017), whose goal is to obtain a global feature representation for each modality and then perform retrieval based on these global features. 2) Fine-grained approaches (Pham et al., 2024; Cheng et al., 2022; Diao et al., 2021; He et al., 2021; Liu et al., 2020; Pan et al., 2023; Zhang et al., 2022) was proposed to establish more detailed correspondences between image and text. Some of these methods (Pham et al., 2024; Cheng et al., 2022; Diao et al., 2021; He et al., 2021; Liu et al., 2020) construct graphs among intra-modal regions or words and aggregate local representations to further capture the semantic relationships between modalities. Despite the progress in recent years, real-world datasets frequently contain noisy correspondences, which inevitably disrupt the alignment process and complicate the accurate measurement of similarity, thereby degrading the overall performance of retrieval models.

### 2.2 NOISY CORRESPONDENCE LEARNING

Noisy correspondence Learning (Huang et al., 2021; Han et al., 2023; Yang et al., 2023; Qin et al., 2023; 2022; Ma et al., 2024; Dang et al., 2024; Yang et al., 2024; Zhao et al., 2024; Feng et al., 2023; Zha et al., 2024; Hu et al., 2023; Han et al., 2024; Duan et al., 2024) focused on developing various robust learning strategies that can handle the modality mismatches. Huang et al. (Huang et al., 2021) first identified the noisy correspondence problem and introduced the Noisy Correspondence Rectifier (NCR). NCR and follow-up works (Han et al., 2023; Yang et al., 2023) leverage a small-loss criterion (Li et al., 2020) to split data into clean and noisy subsets, then apply adaptive prediction functions for label correction. Instead of using the small-loss criterion, some works have employed different metrics to measure the uncertainty of image-text pairs, such as geometrical structure consistency (Zhao et al., 2024), equivariant similarity consistency (Yang et al., 2024), and logits energy-guided sample filtration (Dang et al., 2024). Besides, (Qin et al., 2023; Hu et al., 2023; Qin et al., 2022) have tried to build robust loss functions, and (Han et al., 2024; Duan et al., 2024) have attempted to rematch noisy pairs or assign pseudo-labels to mitigate the adverse effects caused by noisy correspondences. Notably, CREAM (Ma et al., 2024) focuses on "Diverse Potential Consistency" in negative pairs and reweights them via static similarity, while DiffNCL targets weakly-noisy positives by capturing dynamic diffusion discrepancy and reconstructing them into pseudo-clean representations through reverse diffusion. Furthermore, research on the noisy correspondence problem has extended to areas including person re-identification (Qin et al., 2024; Li et al., 2025; Zhang et al., 2025) and visual-language pre-training (Huang et al., 2024), which effectively mitigates the negative impacts of correspondence noise through strategies like sample selection, robust loss design, and graph propagation. In summary, existing research overlooks weakly-noisy correspondences, leading to over-exclusion of informative pairs and under-alignment.

### 2.3 DIFFUSION-BASED MODELS

Diffusion models (Jascha et al., 2015; Ho et al., 2020; Austin et al., 2021; Dhariwal & Nichol, 2021; Park et al., 2024; Kang et al., 2024; Jin et al., 2023; Li et al., 2024) have emerged as a powerful paradigm in generative modeling, characterized by a unique two-stage training process: a forward diffusion process that gradually corrupts the data with additive noise and a backward denoising process that reconstructs the original data through iterative refinement learning. Based on nonequilibrium thermodynamics, these models approximate the data distribution by gradually removing the injected noise through Markov chain transitions. Traditional diffusion methods (e.g., DDPM (Ho et al., 2020)) primarily target unimodal data generation, making it difficult to migrate to cross-modal retrieval tasks directly. Recent cross-modal works like DiffusionRet (Jin et al., 2023) and CUMDR (Li et al., 2024) adapt diffusion models to text-video retrieval and text-based person retrieval by designing denoising networks to learn joint distributions. Despite considerable promise, diffusion models remain scarcely applied to mitigating noisy correspondences in cross-modal retrieval.

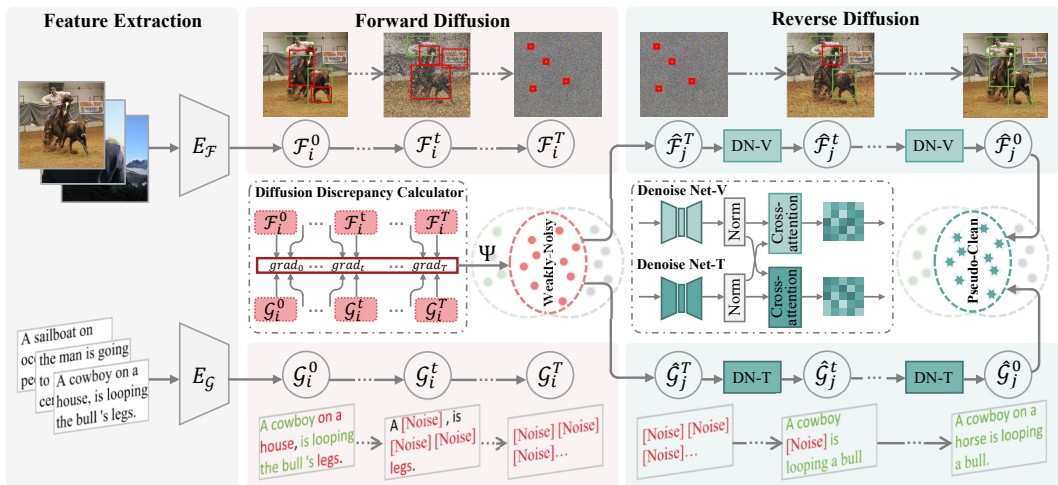

Figure 2: Illustration of the proposed DiffNCL, which employs two main components, i.e., **diffusion forward for weakly-noisy correspondence identification** via synchronized noise injection and diffusion discrepancy calculation, and **diffusion reverse for pseudo-clean representation reconstruction** through modality-specific denoising networks and intra/cross-modal consistency constraints.

## 3 METHODOLOGY

### 3.1 PROBLEM STATEMENT

Technically, consider a training dataset $\mathcal{D} = \{(\mathcal{I}_i, \mathcal{T}_i), y_i\}_{i=1}^N$, where $N$ denotes the data size, $(\mathcal{I}_i, \mathcal{T}_i)$ represents an image-text pair, and $y_i \in \{0, 1\}$ indicates whether the pair belongs to the same instance. The objective of the cross-modal retrieval task is to establish associations between image and text in an unlabeled test set. Under noisy correspondence scenarios, an unknown subset of $\mathcal{D}$ contains mismatched pairs where $(\mathcal{I}_i, \mathcal{T}_i)$ is inherently negative but erroneously labelled as $y_i = 1$. Beyond the widely recognised noisy correspondence problem, an easily overlooked weakly-noisy correspondence phenomenon can also degrade model performance. To mathematically formulate the weakly-noisy correspondence, the semantic associations and atomic semantic units are first defined as follows:

**Definition 1.** *Let the visual modality feature space be $\mathcal{V}$ and the language modality feature space be $\mathcal{L}$. For any $(v, l) \in \mathcal{V} \times \mathcal{L}$, define the semantic association function*

$$\delta : \mathcal{V} \times \mathcal{L} \to \{0, 1\}, \tag{1}$$

*where $\delta(v, l) = 1$ denotes $v$ and $l$ are semantically associated, and $\delta(v, l) = 0$ indicates their semantic disconnection.*

**Definition 2.** *The visual and language atomic unit set $V = \{v_i\}_{i=0}^{K_1}$ and $L = \{l_j\}_{j=0}^{K_2}$ constitutes a cross-modal pair $(V, L)$, whose association structure is defined by the association matrix as follows:*

$$M = [\delta(v_i, l_j)]_{K_1 \times K_2} \in \{0, 1\}^{K_1 \times K_2}. \tag{2}$$

Finally, the mathematical definition of clean, weakly-noisy (abbreviated as "weak" in the formula), and noisy correspondences is given as Definition 3.

**Definition 3.** *For any data pair $(V, L)$, Define the strength of its semantic association:*

$$\rho = \frac{1}{K_1 K_2} \sum_{i=1}^{K_1} \sum_{i=1}^{K_2} \delta(v_i, l_j),$$

$$(V, L) = \begin{cases} clean \iff 1 \geq \rho \geq Max(\frac{1}{K_1}, \frac{1}{K_2}) \iff \forall i, \exists j, \delta(v_i, l_j) = 1 \text{ and } \forall j, \exists i, \delta(v_i, l_j) = 1, \\ weak \iff Max(\frac{1}{K_1}, \frac{1}{K_2}) > \rho > 0 \iff \exists i, \forall j, \delta(v_i, l_j) = 0 \text{ and } \exists j, \forall i, \delta(v_i, l_j) = 0, \\ noisy \iff \rho = 0 \iff \forall (i, j), \delta(v_i, l_j) = 0. \end{cases}$$

$$\tag{3}$$

**Analysis:** Through the interplay of existential and universal quantifiers, Definition 3 rigorously defines the necessary and sufficient conditions for complete semantic alignment and misalignment. Specifically, clean correspondence requires that every visual atomic unit is associated with at least one linguistic atomic unit, and vice versa, ensuring no isolated units in visual and linguistic semantics. Noisy correspondence is defined as atomic units of all modalities being completely unrelated, corresponding to entirely mismatched noise pairs in practice. For the weakly-noisy correspondence, at least one visual or linguistic unit is fully dissociated from all units of the other modality. Notably, $\rho$ serves as a global average indicator of cross-modal atomic unit correlation. The threshold $Max(\frac{1}{K_1}, \frac{1}{K_2})$ of $\rho$ acts as the critical dividing point between clean and weakly-noisy correspondence, determined by the reciprocal maximum number of atomic units in the two modalities, and essentially represents the minimum association density for complete cross-modal semantic alignment.

Due to the excellent performance of (Lee et al., 2018; Anderson et al., 2018), $V$ and $L$ can be regarded as the feature representations $\mathcal{F}_i$ and $\mathcal{G}_i$ by projecting image and text into a shared space via two modality-specific encoders $E_\mathcal{F}$ and $E_\mathcal{G}$ respectively, i.e., $\mathcal{F}_i = E_\mathcal{F}(\mathcal{I}_i)$, $\mathcal{G}_i = E_\mathcal{G}(\mathcal{T}_i)$. Their pairwise similarity $S(\mathcal{F}_i, \mathcal{G}_i)$ is measured by the similarity reasoning networks. To address the weakly-noisy correspondence issue, we propose the DiffNCL approach, as visualized in Figure. 2, to achieve robust cross-modal alignment.

## 3.2 FORWARD DIFFUSION

To effectively distinguish weakly-noisy correspondence samples, the forward diffusion stage captures the inherent discrepancies in image-text pairs with different matching degrees in the diffusion flow.

**Synchronized noise injection.** Inspired by the practice of previous diffusion models (Ho et al., 2020), with the modality-specific noise scheduling implemented over $T$ diffusion steps, synchronized Gaussian noises are first injected into visual features $\mathcal{F}_i$, formulated as:

$$\{\mathcal{F}_i^t\}_{t=1}^T, \mathcal{F}_i^t = \sqrt{\alpha_t}\mathcal{F}_i^{t-1} + \sqrt{1-\alpha_t}\epsilon_1, \tag{4}$$

where $\mathcal{F}_i^0 = \mathcal{F}_i$ represents the original visual feature, and the noise $\epsilon_1 \sim \mathcal{N}(0, I)$ is a random normal vector following the standard Gaussian distribution. The noise scheduling parameter follows $\alpha_t = \cos^2(\frac{\pi t}{2T})$, which ensures that less noise is added during early diffusion steps, with more noise gradually introduced as $t$ increases. Such a design helps reveal latent semantic variations within visual features by adapting the noise level to highlight evolving structural-semantic relationships across diffusion stages. Similarly, for the textual feature, the noise injection formula is:

$$\{\mathcal{G}_i^t\}_{t=1}^T, \mathcal{G}_i^t = \sqrt{\beta_t}\mathcal{G}_i^{t-1} + \sqrt{1-\beta_t}\epsilon_2, \tag{5}$$

where $\mathcal{G}_i^0 = \mathcal{G}_i$, $\epsilon_1 \sim \mathcal{N}(0, I)$, and the noise scheduling parameter for the text modality follows $\beta_t = \cos^3(\frac{\pi t}{2T})$. The difference in the power of the cosine function for $\alpha_t$ and $\beta_t$ is to account for the different characteristics of visual and textual data. Since textual data is more sensitive to noise (Qiu et al., 2022), the cubic-power cosine function for $\beta_t$ results in a slower noise-increasing rate, which helps prevent over-corruption of the semantic information in the text.

**Diffusion discrepancy calculator.** Drawing inspiration from prior works (Sokolić et al., 2017; Fawzi et al., 2018; Ilyas et al., 2019), we posit that sample pairs with varying matching degrees exhibit divergent similarity trajectories during progressive noising. For a series of noised features $\{\mathcal{F}_i^t, \mathcal{G}_i^t\}_{t=1}^T$, the diffusion discrepancy $\Psi_i$ for an image-text pair $(\mathcal{I}_i, \mathcal{T}_i)$ is defined to measure the semantic alignment confidence between image-text pairs, i.e.,

$$\Psi_i = \sum_{t=1}^T \gamma_t \left\| \frac{\partial <\mathcal{F}_i^t, \mathcal{G}_i^t>}{\partial t} \right\|_2^2, \tag{6}$$

where $< \cdot, \cdot >$ denotes cosine similarity function, and $\gamma_t = \frac{(1-\alpha_t)\cdot(1-\beta_t)}{\sum_{t'=1}^T (1-\alpha_{t'})\cdot(1-\beta_{t'})}$ serves as a normalization factor, weighting the contribution of each diffusion step. This metric effectively discriminates clean, weakly-noisy, and noisy samples by quantifying step-wise similarity variations in cross-modal features within the diffusion flow.

**Analysis:** For clean samples, the robust features sustaining semantic consistency between modalities lead to a smaller Jacobian spectral norm (Sokolić et al., 2017), resulting in gentle similarity gradients in the diffusion process and a lower cumulative value $\Psi_i$. In contrast, non-robust features in noisy

samples lack semantic constraints, causing significant fluctuations in similarity gradients upon noise injection and yielding a higher $\Psi_i$, which aligns with the theory in unimodal scenarios that "non-robust features are sensitive to perturbations" (Ilyas et al., 2019). For weakly-noisy samples, some semantically irrelevant features lie in high-curvature regions of the decision boundary (model-sensitive directions) (Fawzi et al., 2018). As noise is incrementally injected via modality-adaptive scheduling, once the noise intensity surpasses their sensitivity threshold, similarity gradients surge at specific steps due to complex local geometric structures, producing $\Psi_i$ values between the extremes. This design of diffusion discrepancies effectively captures the dynamic differences among sample types during diffusion, providing a theoretical analysis for the effective measurement of clean, weakly-noisy, and noisy correspondence.

**Data partitioning.** To effectively identify weakly-noisy correspondences, we propose a hybrid feature representation $\mathcal{H}_i$ combining both sample-wise InfoNCE loss $\ell_i$ and the aforementioned diffusion discrepancy $\Psi_i$, rather than relying solely on the memorization effect, expressed as:

$$\mathcal{H}_i = [\ell_i, \zeta \cdot \Psi_i], \tag{7}$$

where $\zeta = \frac{1}{2}(\mathbb{E}[\sigma(-\ell_i^A)] + \mathbb{E}[\sigma(-\ell_i^B)])$ serves as dynamic weight for regulating the influence of $\Psi_i$. Here, $\mathbb{E}$ and $\sigma(\cdot)$ denote the expectation and sigmoid function, respectively. The $\ell_i$ is defined as:

$$
\begin{aligned}
\ell_i = \ell_{\text{info}}(\mathcal{F}_i, \mathcal{G}_i) = & - \log \frac{\exp(S(\mathcal{F}_i, \mathcal{G}_i)/\tau)}{\exp(S(\mathcal{F}_i, \mathcal{G}_i)/\tau) + \sum_{j \neq i}^N \exp(S(\mathcal{F}_i, \mathcal{G}_j)/\tau)} \\
& - \log \frac{\exp(S(\mathcal{F}_i, \mathcal{G}_i)/\tau)}{\exp(S(\mathcal{F}_i, \mathcal{G}_i)/\tau) + \sum_{j \neq i}^N \exp(S(\mathcal{F}_j, \mathcal{G}_i)/\tau)}
\end{aligned}
\tag{8}
$$

Next, we fit the hybrid features of all training data by using a three-component Gaussian Mixture Model (GMM), modeling the probability distributions of clean, weakly-noisy, and noisy samples, i.e.,

$$p(\mathcal{H}_i|\theta) = \sum_{k=1}^K \xi_k \phi(\mathcal{H}_i|\mu_k, \Sigma_k), \tag{9}$$

where $\xi_k$, satisfying $\sum \xi_k = 1$, represents the mixture coefficient, $\phi(\mathcal{H}_i|k)$ is the probability density of the $k$-th component, and $K = 3$ is set to divide samples into three groups. To avoid self-reinforcing errors and error accumulation, we adopt a co-training paradigm with consensus division. The posterior probability of the $i$-th pair belonging to the clean set is calculated as:

$$P_i^A = \frac{\xi_c \phi(\mathcal{H}_i^A|\mu_c, \Sigma_c)}{\sum_k^K \xi_k^A \phi(\mathcal{H}_i^A|\mu_k^A, \Sigma_k^A)}, \; P_i^B = \frac{\xi_c \phi(\mathcal{H}_i^B|\mu_c, \Sigma_c)}{\sum_k^K \xi_k^B \phi(\mathcal{H}_i^B|\mu_k^B, \Sigma_k^B)}, \tag{10}$$

where the superscripts $A$ and $B$ represent the corresponding models in co-training, and subscript $c$ indicates the clean component of GMM. Through a consensus mechanism of the dual model prediction results, samples are divided into three categories, defined by mask matrices $M_i^c, M_i^w, M_i^n$ to indicate whether the $i$-th sample belongs to the clean, weakly-noisy, or noisy set:

$$
\begin{aligned}
M_i^c &= (\arg\max P_i^A = k_c^A) \wedge (\arg\max P_i^B = k_c^B), \\
M_i^n &= (\arg\max P_i^A = k_n^A) \wedge (\arg\max P_i^B = k_n^B), M_i^w = \neg(M_i^c \vee M_i^n),
\end{aligned}
\tag{11}
$$

where $k_c = \arg\min_k \mu_k$, $k_n = \arg\max_k \mu_k$, and the remaining $k_w$ are the corresponding clean, noisy, and weakly-noisy components of GMM.

### 3.3 Reverse Diffusion

Given a batch of features $\mathcal{B} = \{\mathcal{F}_i^T, \mathcal{G}_i^T | M_i^w = 1\}_{i=1}^B$ with $T$-step noised and $B$ batch size, reverse diffusion aims to reconstruct the semantic correlation features through a series of denoising steps.

**Modality-specific denoising.** Aiming to recover the salient areas of features and eliminate most noise, a series of bottleneck-structured mapping networks $\mathcal{M}_\mathcal{F} = \{\mathcal{M}_\mathcal{F}^t\}_{t=1}^T$ and $\mathcal{M}_\mathcal{G} = \{\mathcal{M}_\mathcal{G}^t\}_{t=1}^T$ are designed to project cross-modal features into a more compact representation space:

$$
\begin{aligned}
\mathcal{M}_\mathcal{F}^t(\hat{\mathcal{F}}_i^{t-1}; \theta) &= \text{LN}\left(\hat{\mathcal{F}}_i^{t-1} + W_\downarrow^t \text{ReLU}(W_\uparrow^t \cdot \hat{\mathcal{F}}_i^{t-1})\right), \\
\mathcal{M}_\mathcal{G}^t(\hat{\mathcal{G}}_i^{t-1}; \theta) &= \text{LN}\left(\hat{\mathcal{G}}_i^{t-1} + W_\downarrow^t \text{ReLU}(W_\uparrow^t \cdot \hat{\mathcal{G}}_i^{t-1})\right),
\end{aligned}
\tag{12}
$$

where $\theta$ denotes the parameters of the projection networks, LN represents the layer normalization, $W_{\downarrow}^t \in \mathbb{R}^{d \times h}$ and $W_{\uparrow}^t \in \mathbb{R}^{h \times d}$ ($h < d$) are the dimensionality-reduction and -expansion projection matrices, respectively, forming a bottleneck structure. Additionally, modality-specific cross-model attention is employed to reconstruct cross-modal association semantics, ensuring that the final denoised features contain only clean correspondences, which can be expressed as:

$$\hat{\mathcal{F}}_i^t = \mathcal{M}_{\mathcal{F}}^t(\hat{\mathcal{F}}_i^{t-1}) + \rho_1 \cdot \text{softmax}\left(\frac{Q(\mathcal{M}_{\mathcal{F}}^t(\hat{\mathcal{F}}_i^{t-1})) \cdot K(\mathcal{M}_{\mathcal{G}}^t(\mathcal{G}_i))}{\sqrt{d}}\right) \cdot V(\mathcal{M}_{\mathcal{G}}^t(\mathcal{G}_i)),$$

$$\hat{\mathcal{G}}_i^t = \mathcal{M}_{\mathcal{G}}^t(\hat{\mathcal{G}}_i^{t-1}) + \rho_2 \cdot \text{softmax}\left(\frac{Q(\mathcal{M}_{\mathcal{G}}^t(\hat{\mathcal{G}}_i^{t-1})) \cdot K(\mathcal{M}_{\mathcal{F}}^t(\mathcal{F}_i))}{\sqrt{d}}\right) \cdot V(\mathcal{M}_{\mathcal{F}}^t(\mathcal{F}_i)),$$

(13)

where $Q, K, V$ are linear projections, and $\rho_1, \rho_2$ are learnable scaling factors. Notably, once the denoising network is sufficiently trained, the final denoised outputs $\hat{\mathcal{F}}_i = \hat{\mathcal{F}}_i^0$ and $\hat{\mathcal{G}}_i = \hat{\mathcal{G}}_i^0$ can be utilized as the pseudo-clean representations to participate in subsequent model training.

**Intra-modal structure consistency.** The intra-structure consistency loss preserves the intrinsic discriminative structure of each modality by enforcing feature reconstruction between the original and denoised representations by element-wise $L_2$ constraints, formulated as:

$$\mathcal{L}_{\text{intra}} = \frac{1}{B} \sum_{i=1}^{B} \left\| \hat{\mathcal{F}}_i - \mathcal{F}_i \right\|_2^2 + \frac{1}{B} \sum_{i=1}^{B} \left\| \hat{\mathcal{G}}_i - \mathcal{G}_i \right\|_2^2. \tag{14}$$

Minimizing this loss ensures that the denoising process retains modality-specific structural information, preventing over-alignment that could erase critical intra-modal discriminative patterns.

**Cross-modal semantic consistency.** Aiming to align the denoised features in the semantic space, the cross-semantic consistency objective employs a contrastive learning framework, which encourages the model to associate reconstructed features with their corresponding pairs while distinguishing them from non-matching instances:

$$\mathcal{L}_{\text{cross}} = -\frac{1}{B} \sum_{i=1}^{B} \log \frac{\exp(<\hat{\mathcal{F}}_i, \hat{\mathcal{G}}_i> /\tau)}{\sum_{j=1}^{B} \left( \exp(<\hat{\mathcal{F}}_i, \mathcal{G}_j> /\tau) + \exp(<\mathcal{F}_j, \hat{\mathcal{G}}_i> /\tau) \right)}. \tag{15}$$

Specifically, the numerator strengthens the similarity of the target pair via exponential operation, treating the reconstructed $(\hat{\mathcal{F}}_i, \hat{\mathcal{G}}_i)$ pair as a pseudo-clean instance to be pulled closer. The denominator is designed to prevent the reconstructed feature $\hat{\mathcal{F}}_j$ from mismatching other original text features $\{\mathcal{G}_k\}_{k=1}^B$ and to prevent the reconstructed text feature $\hat{\mathcal{G}}_j$ from mismatching other original image features $\{\mathcal{F}_k\}_{k=1}^B$. The overall consistency objective combines intra- and cross-modal losses:

$$\mathcal{L}_{\text{consistency}} = \mathcal{L}_{\text{intra}} + \mathcal{L}_{\text{cross}}. \tag{16}$$

## 3.4 ROBUST CROSS-MODAL RETRIEVAL

Furthermore, we also propose a robust contrastive loss $L_{\text{robust}}$, innovatively leveraging the visual and textual pseudo-clean features $\hat{\mathcal{F}}$ and $\hat{\mathcal{G}}$ obtained from diffusion reverse for robust cross-modal retrieval learning. This loss eliminates the interference of noisy correspondences, formulated as:

$$\mathcal{L}_{\text{robust}} = \frac{1}{2B} \sum_{i=0}^{B} \left( \ell_{\text{info}}(\hat{\mathcal{F}}_i, \mathcal{G}_i) + \ell_{\text{info}}(\mathcal{F}_i, \hat{\mathcal{G}}_i) \right), \tag{17}$$

where $\ell_{\text{info}}(\cdot)$ defined in Equation 8, and $B$ denotes the batch size. Based on the above analyses, the comprehensive training objective of our proposed method encompasses a combination of robust cross-modal retrieval loss and diffusion consistency loss, i.e.,

$$\mathcal{L}_{\text{total}} = \mathcal{L}_{\text{robust}} + \mathcal{L}_{\text{consistency}}. \tag{18}$$

## 4 EXPERIMENTS

### 4.1 DATASETS AND METRICS

Following previous studies (Huang et al., 2021), three widely used benchmark datasets, i.e., Flickr30K (Young et al., 2014), MS COCO (Lin et al., 2014), and Conceptual Captions (Sharma

Table 1: Experiment results on CC152K and Flickr30K, where Flickr30K dataset contains 50% weakly-noisy and 40% noisy correspondence. The best results are marked in **bold**.

| Methods | CC152K Image→Text | | | CC152K Text→Image | | | | Flickr30K Image→Text | | | Flickr30K Text→Image | | | |
|---|---|---|---|---|---|---|---|---|---|---|---|---|---|---|
| | R@1 | R@5 | R@10 | R@1 | R@5 | R@10 | rSum | R@1 | R@5 | R@10 | R@1 | R@5 | R@10 | rSum |
| SCAN[ECCV'18] | 30.5 | 55.3 | 65.3 | 26.9 | 53.0 | 64.7 | 295.7 | 36.3 | 69.3 | 80.5 | 24.4 | 54.1 | 67.0 | 331.6 |
| SGR[AAAI'21] | 11.3 | 29.7 | 39.6 | 13.1 | 30.1 | 41.6 | 165.4 | 15.2 | 28.7 | 36.4 | 32.1 | 29.8 | 43.3 | 185.5 |
| NCR[NIPS'21] | 39.5 | 64.5 | 73.5 | 40.3 | 64.6 | 73.2 | 355.6 | 42.3 | 71.1 | 82.3 | 31.0 | 59.0 | 70.7 | 356.4 |
| DECL[MM'22] | 36.2 | 63.6 | 73.2 | 37.1 | 63.6 | 73.7 | 347.4 | 59.3 | 84.8 | 90.9 | 42.3 | 69.0 | 78.3 | 424.7 |
| RCL[TPAMI'23] | 38.3 | 63.0 | 70.4 | 39.2 | 63.2 | 72.3 | 346.4 | 58.9 | 84.7 | 89.8 | 39.5 | 64.1 | 73.5 | 400.5 |
| BiCro[CVPR'23] | 39.7 | 64.6 | 72.6 | 39.2 | 65.0 | 74.1 | 355.2 | 59.1 | 82.8 | 89.1 | 40.4 | 67.7 | 76.6 | 415.7 |
| L2RM[CVPR'24] | 39.5 | 66.2 | 76.0 | 41.8 | 65.9 | 74.9 | 364.3 | 59.9 | 85.6 | 91.2 | 43.8 | 70.4 | 79.9 | 430.8 |
| **DiffNCL** | **40.7** | **68.3** | **77.4** | **42.8** | **68.9** | **76.6** | **374.7** | **67.6** | **88.9** | **94.1** | **47.3** | **74.3** | **83.0** | **455.2** |

Table 2: Experiment results on Flickr30K and MS-COCO. The best results are marked in **bold**.

| Noise | Methods | Flickr30K Image→Text | | | Flickr30K Text→Image | | | | MS-COCO Image→Text | | | MS-COCO Text→Image | | | |
|---|---|---|---|---|---|---|---|---|---|---|---|---|---|---|---|
| | | R@1 | R@5 | R@10 | R@1 | R@5 | R@10 | rSum | R@1 | R@5 | R@10 | R@1 | R@5 | R@10 | rSum |
| 20% | SCAN[ECCV'18] | 58.5 | 81.0 | 90.8 | 35.5 | 65.0 | 75.2 | 406.0 | 62.2 | 90.0 | 96.1 | 46.2 | 80.8 | 89.2 | 464.5 |
| | SGR[AAAI'21] | 55.9 | 81.5 | 88.9 | 40.2 | 66.8 | 75.3 | 408.6 | 25.7 | 58.8 | 75.1 | 23.5 | 58.9 | 75.1 | 317.1 |
| | NCR[NIPS'21] | 75.0 | 93.9 | **97.5** | 58.3 | 83.0 | 89.0 | 496.7 | 76.6 | 95.6 | 98.2 | **62.5** | 89.3 | 95.3 | 517.5 |
| | DECL[MM'22] | 74.5 | 92.9 | 97.1 | 53.6 | 79.5 | 86.8 | 484.4 | 75.6 | 95.1 | 98.3 | 59.9 | 88.3 | 94.7 | 511.9 |
| | RCL[TPAMI'23] | 74.2 | 91.8 | 96.9 | 55.6 | 81.2 | 87.5 | 487.2 | 77.0 | 95.5 | 98.1 | 61.3 | 88.8 | 94.8 | 515.5 |
| | BiCro[CVPR'23] | 76.5 | 93.1 | 97.4 | 58.1 | 82.3 | 88.5 | 495.9 | 76.6 | 95.4 | 98.2 | 61.3 | 88.8 | 94.8 | 515.1 |
| | L2RM[CVPR'24] | 76.5 | 93.7 | 97.3 | 55.5 | 81.5 | 88.0 | 492.5 | **78.4** | 95.7 | 98.3 | 62.1 | 89.1 | 94.9 | 518.5 |
| | **DiffNCL** | **77.4** | **93.8** | 96.8 | **58.5** | **83.4** | **89.5** | **499.4** | 77.6 | **96.1** | **98.5** | 62.2 | **89.7** | **95.4** | **519.5** |
| 40% | SCAN[ECCV'18] | 26.0 | 57.4 | 71.8 | 17.8 | 40.5 | 51.4 | 264.9 | 42.9 | 74.6 | 85.1 | 24.2 | 52.6 | 63.8 | 343.2 |
| | SGR[AAAI'21] | 4.1 | 16.6 | 24.1 | 4.1 | 13.2 | 19.7 | 81.8 | 1.3 | 3.7 | 6.3 | 0.5 | 2.5 | 4.1 | 18.4 |
| | NCR[NIPS'21] | 68.1 | 89.2 | 94.8 | 51.4 | 78.4 | 84.8 | 467.4 | 76.6 | 95.6 | 98.2 | 61.0 | 88.9 | 94.9 | 515.2 |
| | DECL[MM'22] | 72.7 | 92.3 | 95.4 | 53.4 | 79.4 | 86.4 | 479.6 | 75.6 | 95.5 | 98.3 | 59.5 | 88.3 | 94.8 | 512.0 |
| | RCL[TPAMI'23] | 71.3 | 91.1 | 95.3 | 51.4 | 78.0 | 85.2 | 472.3 | 73.9 | 94.9 | 97.9 | 59.0 | 87.4 | 93.9 | 507.0 |
| | BiCro[CVPR'23] | 74.6 | 92.7 | 96.2 | 55.5 | 81.1 | 87.4 | 487.5 | 75.1 | **95.9** | 98.3 | 59.8 | **89.1** | 94.9 | 513.1 |
| | L2RM[CVPR'24] | **75.8** | **93.2** | **96.9** | 56.3 | 81.0 | 87.3 | 490.5 | 75.2 | 94.8 | 98.1 | 59.4 | 87.8 | 94.1 | 509.4 |
| | **DiffNCL** | 75.7 | 92.6 | **96.9** | **56.7** | **82.0** | **88.3** | **492.3** | **76.8** | 95.1 | **98.4** | **61.2** | 89.0 | **95.2** | **515.7** |
| 60% | SCAN[ECCV'18] | 13.6 | 36.5 | 50.3 | 4.8 | 13.6 | 19.8 | 138.6 | 29.9 | 60.9 | 74.8 | 0.9 | 2.4 | 4.1 | 173.0 |
| | SGR[AAAI'21] | 1.5 | 6.6 | 9.6 | 0.3 | 2.3 | 4.2 | 24.5 | 0.1 | 0.6 | 1.0 | 0.1 | 0.5 | 1.1 | 3.4 |
| | NCR[NIPS'21] | 13.9 | 37.7 | 55.5 | 11.0 | 30.1 | 41.4 | 184.6 | 0.1 | 0.3 | 0.4 | 0.5 | 1.0 | 1.0 | 2.4 |
| | DECL[MM'22] | 65.2 | 88.4 | 94.0 | 46.8 | 74.0 | 82.2 | 450.6 | 73.0 | 94.2 | 97.9 | 57.0 | 86.6 | 93.8 | 502.5 |
| | RCL[TPAMI'23] | 71.3 | **91.1** | 95.3 | 51.4 | 78.0 | 85.2 | 472.3 | 73.9 | **94.9** | 97.9 | 59.0 | 87.4 | 93.9 | 507.0 |
| | BiCro[CVPR'23] | 67.6 | 90.8 | 94.4 | 51.2 | 77.6 | 84.7 | 466.3 | 73.9 | 94.7 | 97.9 | 58.7 | 87.0 | 93.8 | 506.0 |
| | L2RM[CVPR'24] | 70.0 | 90.8 | 95.4 | 51.3 | 76.4 | 83.7 | 467.6 | **75.4** | 94.7 | 97.9 | 59.2 | 87.4 | 93.8 | 508.4 |
| | **DiffNCL** | **71.7** | 90.0 | **95.5** | **53.0** | **78.6** | **86.0** | **474.8** | 74.9 | **94.9** | **98.1** | **59.5** | **87.8** | **94.5** | **509.7** |

et al., 2018), are introduced in the experiments. Detailed descriptions are given in the Appendix. For evaluation, the recall at K (R@K) metric is used to evaluate the retrieval performance. Specifically, R@K measures the proportion of relevant items retrieved from the top K results. In our experiments, we report R@1, R@5, R@10 results of image-to-ext and text-to-image retrieval. The sum of these three recalls, i.e., rSum, is utilized to evaluate the overall performance following (Huang et al., 2021).

### 4.2 COMPARISON WITH STATE-OF-THE-ARTS

In our experiments, we conduct a comprehensive comparison with the state-of-the-art methods, including SCAN (Lee et al., 2018), NCR (Huang et al., 2021), DECL (Qin et al., 2022), RCL (Hu et al., 2023), BiCro (Yang et al., 2023), and L2RM (Han et al., 2024). To ensure a fair comparison, the SGR model is adopted as the backbone in the compared methods.

**Evaluation on Real-World Noisy Correspondence.** Quantitative results from evaluations on the CC152K dataset are reported to validate scenarios involving real-world noisy correspondences. As shown in Table 1, DiffNCL outperforms baseline models by a considerable margin, achieving an overall rSum with a 10.4% performance improvement compared to the second-best L2RM.

Table 3: Ablation studies on Flickr30K with 20% noise. w/ denotes "with".

| Method | Image→Text | | | Text→Image | | | rSum |
|---|---|---|---|---|---|---|---|
| | R@1 | R@5 | R@10 | R@1 | R@5 | R@10 | |
| Base | 75.3 | 93.0 | 97.1 | 57.3 | 82.9 | 88.9 | 494.6 |
| Base w/ FD | 76.0 | 93.2 | 96.7 | 57.6 | 83.0 | 89.1 | 495.6 |
| Base w/ RD | 76.2 | 93.7 | **97.6** | 57.6 | 83.3 | 89.4 | 497.8 |
| DiffNCL | **77.4** | **93.8** | 96.8 | **58.5** | **83.4** | **89.5** | **499.4** |

Significantly, our DiffNCL yields an improvement of 1.0% R@1, 2.1% R@5, 1.4% R@10 for image-to-text retrieval, and 1.0% R@1, 3.0% R@5, 1.7% R@10 for text-to-image retrieval than the second-best method, consistently highlighting its robustness and effectiveness in handling real-world noisy correspondence. Compared with synthetic noisy correspondence, our method demonstrates superior adaptability to real-world noise environments, indicating that: i) the weakly-noisy correspondence issue is particularly pronounced under real-world scenarios; ii) DiffNCL effectively mitigates the challenges posed by weakly-noisy correspondences.

**Evaluation on Synthetic Weakly-Noisy Correspondence.** To further study the robustness of the DiffNCL method in the weakly-noisy correspondence environment, we conducted synthetic noise experiments on the Flickr30K dataset with 50% weakly-noisy and 40% noisy correspondence to simulate the complex real-world cross-modal retrieval scenarios. In particular, the weakly-noisy correspondence are generated by randomly replacing several words in a sentence at a specific weakly-noisy ratio. The comparative results are summarized in Table 1. We can observe that all methods suffer from varying degrees of performance degradation under the influence of weakly-noisy data. Nonetheless, the proposed method consistently achieves significant performance compared to all robust baselines. Specifically, our DiffNCL yields an improvement of 7.7% R@1, 3.3% R@5, 2.9% R@10 for image-to-text and 3.5% R@1, 3.9% R@5, 3.1% R@10 for text-to-image retrieval than the second-best method, respectively.

**Evaluation on Synthetic Noisy Correspondence.** We further investigate the robustness of our DiffNCL approach in the synthetic noisy correspondence environment. To analyze the performance and robustness of all baselines under different noise rates, we adopt 20%, 40%, and 60% synthetic noise on the training sets of Flickr30k and MS-COCO to simulate noisy correspondence. For the results of MS-COCO, we report the average on 5 folds of 1K test images. The test results are presented in Table 2. Specifically, on the Flickr30K dataset, DiffNCL achieves an overall rSum with various noise ratios improvement of 7.0%, while on the MS-COCO dataset, the overall rSum increases by 2.8%. This demonstrates that the proposed DiffNCL outperforms robust baselines including NCR, DECL, RCL, BiCro, and L2RM across most evaluation metrics, indicating its superior robustness to the challenge of modal mismatch in cross-modal retrieval. Additionally, comparison results of MS-COCO 5K are provided in the supplementary material.

### 4.3 ABLATION STUDY

To systematically evaluate the contribution of each component, we conduct ablation studies on Flickr30K with 20% synthetic noise. The "Base" variant builds upon NCR (Huang et al., 2021), employing GMM partitioning based solely on $\ell_i$ and robust InfoNCE loss, excluding our diffusion modules. "Base w/ FD" and "Base w/ RD" then incrementally add forward and reverse diffusion stages, respectively, with the full "DiffNCL" integrating both.

**Effect on Forward Diffusion.** To investigate the impact of forward diffusion, we design the variation, i.e., "Base w/ FD", which denotes that (i) the diffusion discrepancy calculator is incorporated to get per-sample diffusion discrepancies, (ii) and feeds these discrepancies to GMM for enhancing the discrimination capability. The comparison results in Table 3 demonstrate that the diffusion forward stage effectively improves the ability to identify different samples, causing a performance improvement of rSum by 1.0%, indicating its enhancement of model robustness.

**Effect on Backward Diffusion.** "Base w/ RD" variant is designed to investigate the impact of reverse diffusion, which means that (i) the modality-specific denosing net with the diffusion consistency loss is introduced to the Base model, (ii) and the denoised pseudo-clean representations are replacing the original noisy features to compute robust cross-modal retrieval loss. The comparison results in Table 3 indicate that introducing the reverse diffusion stage effectively makes a performance

improvement of rSum by 3.2%, revealing that the original noisy correspondences are effectively reconstructed into pseudo-clean correspondences, thereby enhancing the robustness of the model.

## 5 CONCLUSION

In this paper, we presented Diffusion-Driven Weakly-Noisy Correspondence Learning (DiffNCL), the first unified forward–reverse diffusion framework tailored to mitigate weakly-noisy correspondences in cross-modal retrieval. By leveraging a novel forward diffusion mechanism to mine and amplify subtle distributional discrepancies, DiffNCL accurately separates clean, weakly-noisy, and strongly noisy pairs—thereby alleviating both over-exclusion and under-alignment. The reverse diffusion stage further transforms corrupted features into high-fidelity pseudo-clean embeddings under dual consistency constraints, enabling robust cross-modal supervision without discarding informative samples. Our framework not only delivers significant gains in retrieval accuracy and robustness but also opens new avenues for integrating diffusion dynamics into multimodal representation learning.

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

## A  EXPERIMENTAL SETTINGS

### A.1  DATASET DESCRIPTIONS

To validate the effectiveness of our approach, we conduct experiments on three widely used benchmark datasets for cross-modal retrieval, described as follows: Specifically, collected from the Flickr website, Flickr30K contains 31,000 images with 5 corresponding captions. We use 1,000 image-text pairs for validation, 1,000 for testing, and 29,000 for training. MS-COCO includes 123,287 images with five captions each. Following the data partition in (Lee et al., 2018), 5,000 images are used for modal validation, 5,000 for model testing, and the rest 113,287 for model training. Conceptual Captions is a large-scale dataset with 3%~20% real-world correspondence noise. Using a subset of Conceptual Captions named CC152K, which is split by (Huang et al., 2021), contains 150,000 image-text pairs for training, 1,000 pairs for validation, and 1,000 pairs for testing.

### A.2  IMPLEMENTATION DETAILS

The proposed DiffNCL is a general and robust framework that can be easily extended to cross-modal retrieval methods to mitigate noisy correspondence. To ensure fair comparisons, we employed the SGR model as the backbone, with all settings of the main experiments consistent with NCR. Notably, this work does not use pre-existing diffusion models (e.g., DDPM (Ho et al., 2020), DiffusionRet (Jin et al., 2023)); instead, we designed a task-specific forward–reverse diffusion process tailored for cross-modal weakly-noisy correspondence learning. Specifically, the Adam optimizer was exclusively used, with the batch size set to 128 and an initial learning rate of 0.0002. Moreover, all temperature parameters involved in the experiments were fixed at 0.07. To avoid self-reinforcing errors and error accumulation, the co-training strategy was adopted during training. For the Flickr30K dataset, the model underwent 5 warm-up epochs, while 10 warm-up epochs were applied to the COCO and

CC152K datasets. Post-warm-up training epochs were set to 40, 20, and 40 for the Flickr30K, COCO, and CC152K datasets, respectively. During inference, the averaged prediction from models A and B was used. For key components involved in the diffusion process and data partitioning, additional implementation details are supplemented as follows: the Gaussian Mixture Model (GMM) used for sample partitioning (clean, weakly-noisy, noisy) adopts 100 iterations with random initialization, and the covariance regularization parameter $reg\_covar$ is set to $1e-3$; feature dimension settings include the original visual and textual feature dimension $d$ (output by modality-specific encoders $E_{\mathcal{F}}$ and $E_{\mathcal{G}}$) of 1024, and the bottleneck dimension $h$ in the modality-specific denoising networks $\mathcal{M}_{\mathcal{F}}$ and $\mathcal{M}_{\mathcal{F}}$ of 512, ensuring compact representation while preserving discriminative semantic information.

## A.3 TRAINING PIPELINES

---

**Algorithm 1:** The training pipeline of our DiffNCL

---

**Input:** A training cross-modal dataset $\mathcal{D}$, image-text matching model $\mathcal{S}(\theta_1)$, diffusion denoising network $\mathcal{R}(\theta_2)$;
**Output:** Trained models $\mathcal{S}(\theta_1)$ and $\mathcal{R}(\theta_2)$

1 Initialize the training parameters $\theta_1$ and $\theta_2$ and all the hyper-parameters;
2 **for** *each epoch* **do**
3      **for** $\mathcal{F}, \mathcal{G}$ *in* $\mathcal{D}$ **do**
4          **for** $t = 1$ **to** $T$ **do**
5              Add sync Gaussian noise to $\mathcal{F}, \mathcal{G}$;
6              Calculate and aggregate per-step cosine similarity;
7          **end**
8          Obtain the per-sample diffusion discrepancy;
9          Obtain the per-sample loss;
10      **end**
11      Feed discrepancies and losses into 3-component GMM;
12      Split $\mathcal{D}$ into clean subset $\mathcal{D}_c$, wealy-noisy $\mathcal{D}_w$ and noisy subset $\mathcal{D}_n$;
13      **for** $\mathcal{F}, \mathcal{G}$ *in* $\mathcal{D}_c$ **do**
14          Obtain similarities via $\mathcal{S}(\mathcal{F}, \mathcal{G})$;
15          Compute the retrieval loss;
16      **end**
17      **for** $\mathcal{F}, \mathcal{G}$ *in* $\mathcal{D}_w, \mathcal{D}_n$ **do**
18          Reconstruct pseudo-clean features via $\hat{\mathcal{F}}, \hat{\mathcal{G}} = \mathcal{R}(\mathcal{F}, \mathcal{G})$;
19          Obtain similarities via $\mathcal{S}(\hat{\mathcal{F}}, \hat{\mathcal{G}})$;
20          Compute the robust and consistency loss;
21      **end**
22      Obtain overall loss $\mathcal{L}$;
23      $\theta_1, \theta_2 = \text{Optimizer}([\theta_1, \theta_2], \mathcal{L})$
24 **end**

---

## B BROADER EXPERIMENTS

### B.1 COMPUTATIONAL COMPLEXITY

Table 4: Computational results of backbone and diffusion module

| Components | GFLOPs | Parameters(M) | Per Iteration Wall-Clock Time(S) |
|---|---|---|---|
| Backbone | 180.1 | 18.11 | 0.4236 |
| Diffusion Net (Training only) | 123.4 | 8.400 | 0.0273 |

To analyze the computational complexity of our DiffNCL, we conducted quantitative analyses of FLOPs and wall-clock time, as shown in Table 4 and Table 5. Additionally, we report the computational cost under different diffusion steps in Table 6, including forward time, backward time, and peak memory usage. The forward time measures the complete forward propagation from

Table 5: Computational results of different methods

| Methods | Ref. | Parameters(M) | Per Epoch Wall-Clock Time(Minute) |
|---|---|---|---|
| SGR | AAAI'18 | 18.11 | 20.47 |
| NCR (baseline) | NeurIPS'21 | 36.22 | 30.20 |
| DECL-SGRAF | MM'22 | 36.22 | 32.06 |
| DECL-SGR | MM'22 | 18.11 | 17.49 |
| L2RM | CVPR'24 | 18.13 | 29.52 |
| **DiffNCL** | Ours | 42.52 | 38.68 |

Table 6: Computational results of different diffusion steps

| T-step | Per Iteration Forward Time(S) | Per Iteration Backward Time(S) | Peak Memory(MB) |
|---|---|---|---|
| T=3 | 0.1392 | 0.2334 | 1,611.85 |
| T=4* | 0.1389 | 0.2357 | 1,673.34 |
| T=5 | 0.1397 | 0.2384 | 1,724.67 |
| T=7 | 0.1400 | 0.2381 | 1,860.48 |
| T=10 | 0.1483 | 0.2407 | 2,023.62 |
| T=15 | 0.1536 | 0.2452 | 2,341.47 |

input to final loss, including both backbone and diffusion modules. The backward time measures the gradient computation for all parameters, and the "Diffusion Net (Training only)" time in Table 4 refers only to the diffusion-specific operations during training. The peak memory measures the maximum GPU memory usage during a complete training iteration (forward and backward), including model parameters, activations, gradients, and temporary buffers, but excluding optimizer states and system overhead. This measurement focuses on the method-specific memory requirements and enables fair comparison across different configurations.

It's common and well known that the diffusion process introduces additional computational overhead, primarily due to the repeated feature transformations across T steps. To address this, we implemented several optimizations: reducing the diffusion step count to $T = 4$(achieved remarkable performance), adopting parameter sharing in modality-specific denoising networks, and using lightweight bottleneck structures to minimize redundant computations. As shown in Table 6, the computational cost (time and memory) increases with the number of diffusion steps, but our chosen step T=4 strikes a balance between performance and efficiency. The experimental results show that while the diffusion process introduces increases in FLOPs, the actual training wall-clock time increases merely.

## B.2 DETAILED ABLATION STUDY

Table 7: Detailed ablation studies on Flickr30K with 20% noise. w/o denotes "without".

| Method | Image→Text | | | Text→Image | | | |
|---|---|---|---|---|---|---|---|
| | R@1 | R@5 | R@10 | R@1 | R@5 | R@10 | rSum |
| DiffNCL w/o $\mathcal{L}_{intra}$ | 74.4 | 92.6 | 96.0 | 56.8 | 82.2 | 88.3 | 490.3 |
| DiffNCL w/o $\mathcal{L}_{cross}$ | 76.9 | 93.5 | 96.6 | 58.0 | 83.2 | 89.0 | 497.2 |
| DiffNCL | **77.4** | **93.8** | **96.8** | **58.5** | **83.4** | **89.5** | **499.4** |

To further dissect the role of dual consistency constraints in the reverse diffusion stage, we conduct detailed ablation experiments on Flickr30K with 20% synthetic noise. We construct variants by removing each consistency loss (while retaining other DiffNCL components) to isolate the impact of intra-modal structural consistency ($\mathcal{L}_{intra}$) and cross-modal semantic consistency ($\mathcal{L}_{cross}$), with results in Table 7.

**Effect on Intra-modal Structural Consistency ($\mathcal{L}_{intra}$).** The "DiffNCL w/o $\mathcal{L}_{intra}$" variant omits the intra-modal loss, relying only on cross-modal constraints and denoising networks. It achieves an rSum of 490.3 (9.1 lower than full DiffNCL): Image→Text drop by 3.0% R@1, 1.2% R@5,

0.8% R@10, and Text→Image drop by 1.7% R@1, 1.2% R@5, 1.2% R@10. This confirms $\mathcal{L}_{intra}$ preserves modality-specific discriminative topology via $L_2$ constraints between original and denoised features, preventing semantic collapse and maintaining feature discrimination.

**Effect on Cross-modal Semantic Consistency ($\mathcal{L}_{cross}$).** The "DiffNCL w/o $\mathcal{L}_{cross}$" variant removes the cross-modal loss, retaining only intra-modal constraints. Its rSum of 497.2 is 2.2 lower than full DiffNCL: Image→Text decrease by 0.5% R@1, 0.3% R@5, 0.2% R@10, and Text→Image decrease by 0.5% R@1, 0.2% R@5, 0.5% R@10. This shows $\mathcal{L}_{intra}$ drives denoised features toward clean manifolds by enhancing valid semantic alignment and suppressing spurious similarities, complementing intra-modal constraints for cross-modal coherence.

Ablation results validate the synergy of dual constraints: $\mathcal{L}_{intra}$ safeguards intra-modal structural integrity, while $\mathcal{L}_{cross}$ ensures cross-modal semantic alignment. Together, they enable reverse diffusion to generate high-fidelity pseudo-clean features, underpinning DiffNCL's robustness to weakly-noisy correspondences.

### B.3 HYPERPARAMETER SENSITIVITY

We have conducted additional hyperparameter sensitivity experiments, including noise schedule, diffusion step, warm-up epoch, and clustering approach, and provided detailed guidelines for adaptation.

**Analysis of noise schedule.** We systematically evaluated different noise scheduling strategies as shown in the Table 8. The key validation results show that the proposed configuration achieves optimal performance, demonstrating the effectiveness of the modality-specific noise scheduling design. Moreover, the performance remains robust across variations in composition, indicating the stability of our DiffNCL method.

Table 8: Evaluation results of various noise scheduling combinations under 20% noise ratio on Flickr30K dataset. * denotes the configuration we selected.

| Schedule combination | Image→Text | | | Text→Image | | | rSum |
| --- | --- | --- | --- | --- | --- | --- | --- |
| | R@1 | R@5 | R@10 | R@1 | R@5 | R@10 | |
| $\alpha = $ Linear, $\beta = $ Linear | 74.5 | 93.3 | 96.6 | 58.1 | 83.4 | 89.7 | 496.1 |
| $\alpha = \cos^2, \beta = $ Linear | 75.7 | 94.1 | 97.6 | 58.1 | 83.0 | 88.6 | 497.1 |
| $\alpha = $ Linear, $\beta = \cos^3$ | 74.4 | 93.3 | 96.9 | 57.4 | 83.2 | 89.2 | 494.4 |
| $\alpha = \cos^3, \beta = \cos^2$ | 75.1 | 94.1 | 96.9 | 58.2 | 83.2 | 89.7 | 497.2 |
| $\alpha = \cos^2, \beta = \cos^2$ | 74.8 | 93.7 | 97.6 | 58.4 | 83.4 | 89.5 | 497.3 |
| $\alpha = \cos^3, \beta = \cos^3$ | 77.1 | 93.6 | 96.9 | 58.3 | 83.3 | 89.7 | 498.9 |
| $\alpha = \cos^2, \beta = \cos^3$* | 77.4 | 93.8 | 96.8 | 58.5 | 83.4 | 89.5 | 499.4 |

**Analysis of diffusion step.** We evaluated the impact of diffusion steps in Table 9, which suggest that removing the diffusion module led to a significant performance degradation, yielded consistent performance, within which consistent performance is achieved; and $T = 4$ balanced computational cost and effectiveness with good performance.

Table 9: Evaluation results of different diffusion steps under 20% noise ratio on Flickr30K dataset. * denotes the configuration we selected.

| $T$-step | Image→Text | | | Text→Image | | | rSum |
| --- | --- | --- | --- | --- | --- | --- | --- |
| | R@1 | R@5 | R@10 | R@1 | R@5 | R@10 | |
| $T = 0$ | 75.3 | 93.0 | 97.1 | 57.3 | 82.9 | 88.9 | 494.6 |
| $T = 2$ | 76.2 | 94.0 | 96.9 | 58.0 | 83.3 | 89.2 | 496.8 |
| $T = 4$* | 76.6 | 93.9 | 97.6 | 58.5 | 83.0 | 89.4 | 499.1 |
| $T = 8$ | 74.8 | 94.0 | 97.6 | 58.0 | 83.5 | 89.5 | 497.4 |
| $T = 16$ | 75.4 | 94.7 | 97.3 | 58.8 | 83.9 | 90.5 | 500.6 |
| $T = 20$ | 76.0 | 93.1 | 97.5 | 58.3 | 83.7 | 89.9 | 498.5 |

**Analysis of warm-up epoch.** We investigated the impact of warm-up epochs on model convergence in Table 10. Practical guidelines regarding warm-up epochs are as follows: 5 epochs are recommended as they provide an optimal balance between convergence and efficiency, even without warm-up, the method maintains competitive performance, and extended warm-up may lead to slight degradation.

Table 10: Evaluation results of various warm-up epochs under 20% noise ratio on Flickr30K dataset. * denotes the configuration we selected.

| Training warm-up | Image→Text | | | Text→Image | | | rSum |
|---|---|---|---|---|---|---|---|
| | R@1 | R@5 | R@10 | R@1 | R@5 | R@10 | |
| epoch = 0 | 74.3 | 94.1 | 97.5 | 58.4 | 83.2 | 89.1 | 496.6 |
| epoch = 5* | 76.6 | 93.9 | 97.6 | 58.5 | 83.0 | 89.4 | 499.1 |
| epoch = 10 | 76.3 | 93.2 | 97.4 | 58.3 | 83.3 | 89.3 | 497.7 |
| epoch = 15 | 73.8 | 94.5 | 97.2 | 58.2 | 83.0 | 89.3 | 496.0 |

### B.4 BACKBONE GENERALIZATION

To evaluate the generalization of DiffNCL across diverse architectural configurations—including integration with large-scale pre-trained models and adaptation to dedicated cross-modal backbones—we conduct a series of experiments to verify its robustness, adaptability, and noise resilience. The results, supported by Table 11 (MS-COCO 5K) and Table 12 (Flickr30K), demonstrate that DiffNCL maintains superior performance across different architectural setups, validating its architecture-agnostic design.

**Integration with Pre-trained Model CLIP.** We first assess DiffNCL's compatibility with CLIP Radford et al. (2021), a renowned large-scale pre-trained model trained on 400 million web-collected image-text pairs. Experimental results show that CLIP exhibits significant performance drops when fine-tuned with noisy data. For example, CLIP (ViT-B/32) has a zero-shot rSum of 361.6, but this plummets to 236.3 after fine-tuning. Even the larger ViT-L/14 variant sees its fine-tuning rSum drop to 289.4, far below its zero-shot performance (400.4). Importantly, by integrating DiffNCL with CLIP (ViT-B/32) (denoted as "DiffNCL+CLIP"), we observe a dramatic performance boost. On MS-COCO 5K under 20% noise, DiffNCL+CLIP achieves an rSum of 451.8, with Image→Text R@1 (62.7%) and Text→Image R@1 (48.2%) reaching the highest among all variants, highlighting its ability to enhance pre-trained models' resistance to noisy correspondences.

**Adaptation to Dedicated Cross-Modal Backbones.** To further validate DiffNCL's adaptability to specialized cross-modal architectures, we test it with two dedicated backbones (SAF and SGRAF)Diao et al. (2021); Huang et al. (2021) under 60% high noise (a challenging scenario for most methods) on the Flickr30K dataset. Experimental results show that DiffNCL consistently outperforms state-of-the-art methods across both backbones, demonstrating its generality and adaptability. For the SAF backbone, DiffNCL-SAF achieves an rSum of 468.6, surpassing DECL-SAF and BiCro-SAF by 10.2 and 11.6, respectively, while the R@1 metric of 67.9% and 51.6% outperforms the second-best by 1.8% and 3.8%. For the SGRAF backbone, DiffNCL achieves an rSum of 480.6, outperforming L2RM-SGRAF and BiCro-SGRAF by 13.0 and 14.3. Notably, the R@1 metric of DiffNCL leads significant margins by, while the R@1 metric of 71.8% and 54.4% outperforms the second-best by 1.8% and 2.2%. The superior performance of DiffNCL across different backbones validates its architecture-agnostic design, as it effectively enhances noise resilience through integrating diffusion dynamics for weakly-noisy detection and pseudo-clean representation reconstruction, establishing it as a general solution for robust cross-modal retrieval tasks.

Across both large-scale pre-trained models (CLIP) and dedicated cross-modal backbones (SAF, SGRAF), DiffNCL consistently enhances performance—even under high noise levels. Its core advantage lies in leveraging diffusion dynamics to mine weak-noise discrepancies and reconstruct pseudo-clean features, enabling architecture-agnostic noise resilience. This validates DiffNCL's generalization as a universal solution for robust cross-modal retrieval tasks.

### B.5 COMPREHENSIVE WEAKLY-NOISY EXPERIMENTS

Figure 3 demonstrates the results in comprehensive weakly-noisy experiments. As the proportion of weak noise increases from 20% to 50% and the noise ratio rises from 20% to 60%, the model exhibits

Table 11: Experiment results on MS-COCO 5K.

| Noise Ratio | Methods | Image→Text | | | Text→Image | | | rSum |
|---|---|---|---|---|---|---|---|---|
| | | R@1 | R@5 | R@10 | R@1 | R@5 | R@10 | |
| 0%, Zero-Shot | CLIP (ViT-L/14) | 58.4 | 81.5 | 88.1 | 37.8 | 62.4 | 72.2 | 400.4 |
| | CLIP (ViT-B/32) | 50.2 | 74.6 | 83.6 | 30.4 | 56.0 | 66.8 | 361.6 |
| 20%, Fine-tune | CLIP (ViT-L/14) | 36.1 | 61.3 | 72.5 | 22.6 | 43.2 | 53.7 | 289.4 |
| | CLIP (ViT-B/32) | 21.4 | 49.6 | 63.3 | 14.8 | 37.6 | 49.6 | 236.3 |
| | **DiffNCL+CLIP** | **62.7** | **86.4** | **92.8** | **48.2** | **76.1** | **85.3** | **451.8** |

Table 12: Experiment results under 60% noise ratio on Flickr30K.

| Method | Image→Text | | | Text→Image | | | rSum |
|---|---|---|---|---|---|---|---|
| | R@1 | R@5 | R@10 | R@1 | R@5 | R@10 | |
| DECL-SAF | 66.4 | 88.1 | 93.6 | 49.8 | 76.1 | 84.4 | 458.4 |
| BiCro-SAF | 67.1 | 88.3 | 93.8 | 48.8 | 75.2 | 83.8 | 457.0 |
| L2RM-SAF | 66.1 | 88.8 | 93.8 | 47.8 | 74.2 | 82.2 | 452.9 |
| **DiffNCL-SAF** | **67.9** | **90.7** | **95.0** | **51.6** | **77.8** | **85.6** | **468.6** |
| DECL-SGRAF | 69.4 | 89.4 | 95.2 | 52.6 | 78.8 | 85.9 | 471.3 |
| BiCro-SGRAF | 67.6 | 90.8 | 94.4 | 51.2 | 77.6 | 84.7 | 466.3 |
| L2RM-SGRAF | 70.0 | 90.8 | 95.4 | 51.3 | 76.4 | 83.7 | 467.6 |
| **DiffNCL-SGRAF** | **71.8** | **91.5** | **95.5** | **54.4** | **80.2** | **87.2** | **480.6** |

significant robustness for both image-to-text and text-to-image retrieval. Specifically, DiffNCL maintains relative stability in recall rates as the weak-noise proportion increases. Notably, in complex scenarios with 50% weak noise and 60% noise, it still maintains a certain retrieval accuracy. This highlights the robust capabilities in accurately capturing semantic associations and resisting noise under diverse noise of DiffNCL.

## B.6 VISUALIZATION ANALYSIS

To validate the semantic discriminative power of forward diffusion and the semantic restoration efficacy of reverse diffusion, we analyze the box plots of the discrepancy distribution $\Psi_i$. For statistical representativeness and alignment with the main experiment's data scale, we first randomly select 500 samples from each of the three original sample types (Clean, Weakly-noisy, Noisy); the Pseudo-clean samples are then generated by inputting the randomly selected Weakly-noisy samples into the reverse diffusion module for reconstruction, ensuring the Pseudo-clean set directly corresponds to the Weakly-noisy set in terms of sample source.

In forward diffusion, clean samples exhibit low and concentrated $\Psi_i$ values, indicating strong semantic robustness that resists noise-induced perturbations; weakly-noisy samples show medium $\Psi_i$ levels with relatively compact distributions, reflecting conditional sensitivity—partial semantic units respond to specific noise intensities while overall stability is maintained; noisy samples demonstrate significantly higher and wider $\Psi_i$ distributions, revealing inherent semantic fragility that leads to continuous similarity degradation under perturbation. The clear separation of these three distributions directly proves that forward diffusion can systematically distinguish samples with varying semantic robustness, laying a critical foundation for accurate weakly-noisy correspondence identification. Meanwhile, in reverse diffusion, the $\Psi_i$ distribution of pseudo-clean samples shifts remarkably toward that of clean samples, which confirms that the dual consistency constraints (intra-modal structural and cross-modal semantic consistency) effectively repair weakly-noisy and noisy features into robust pseudo-clean representations.

## B.7 CASE STUDY

To further reveal the actual effect of the model in different cross-modal retrieval cases, we visualize several results of the top-5 retrieved instances on the CC152K dataset. As shown in Figure 5, we

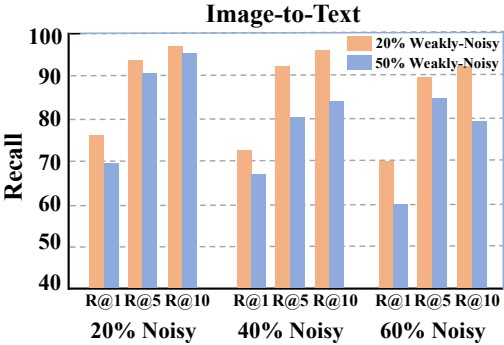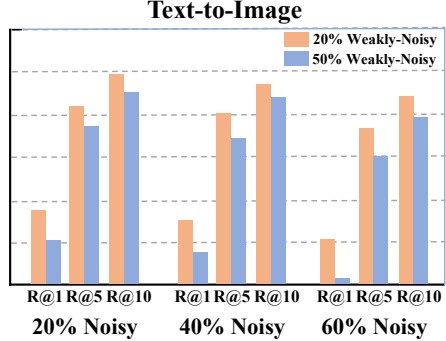

Figure 3: Illustration of experiment results under comprehensive noisy settings.

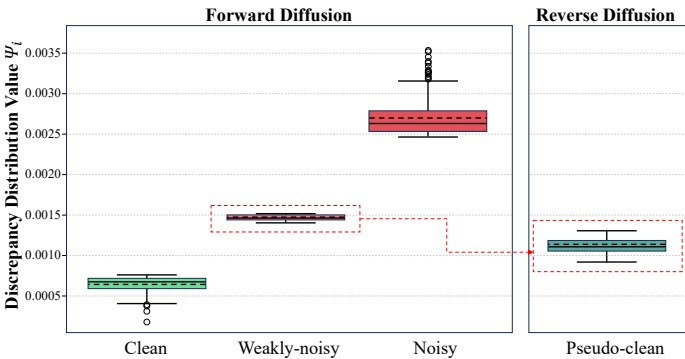

Figure 4: Illustration of the statistical distribution of diffusion discrepancy across clean, weakly-noisy, noisy, and pseudo-clean samples in the forward-reverse diffusion process.

can observe the following conclusions: **(i) Cross-modal retrieval results across diverse scenarios exhibit the model's remarkable performance.** In unambiguous contexts like "people waiting for the bus in a snowstorm" and "a single tropical palm tree...sunny blue sky", the model achieves high GT similarity scores (0.9972 and 0.9875), demonstrating its ability to capture core semantic associations and align features accurately. **(ii) The model maintains robust retrieval stability in cases involving complex multi-element queries**. Non-GT results are ranked by semantic relevance, such as "river", "fields", illustrating its ability to handle composite scenes with multiple visual/textual elements and prioritize relevant features over noise—even when faced with less relevant outliers such as "parking garage". **(iii) Across all scenarios, non-GT results are consistently ordered by semantic relatedness.** Irrelevant entries, such as "fir tree" for a tropical palm query or "automobile industry business" for a parked news car image, receive lower similarity scores. This highlights the model's ability to distinguish and rank cross-modal pairs based on semantic relevance, underscoring its generalizable capacity to organize retrievals by content rather than superficial keyword matches.

## C    THEORETICAL TIME COMPLEXITY OF THE DIFFUSION COMPONENTS

In this section we provide a formal complexity analysis of the core diffusion mechanisms used in DiffNCL, including the forward diffusion (noising) module and the reverse diffusion (denoising) module. All complexities are measured *per training batch* of size $B$, and Big-$O$ notation hides constant factors and fixed hyperparameters. The following symbols are used throughout:

### C.1    FORWARD DIFFUSION COMPLEXITY

**Proposition C.1** (Forward diffusion complexity)**.** *The forward diffusion module has time complexity* $O(Td)$ *per batch and parameter complexity* $O(1)$.

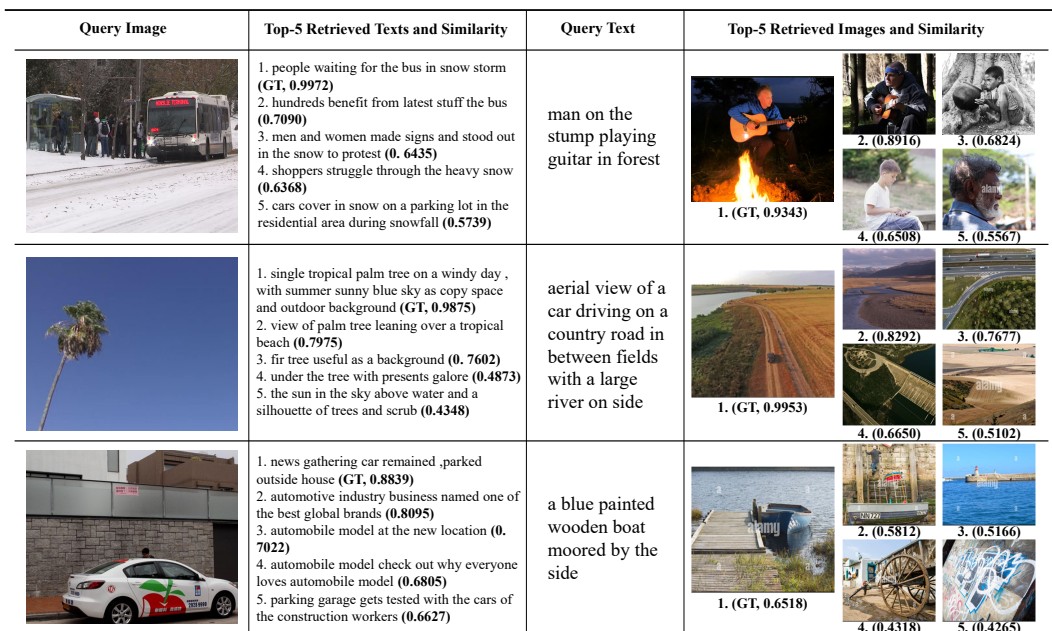

Figure 5: Illustration of Top-10 returned results for cross-modal retrieval. The pair-wise similarity is in brackets, and "GT" denotes the ground-truth.

| Symbol | Description |
|---|---|
| $d$ | Dimensionality of the joint embedding space |
| $h$ | Bottleneck dimensionality in the denoising network ($h < d$) |
| $T$ | Number of diffusion steps |
| $B$ | Batch size (fixed) |
| $K$ | Number of GMM components (fixed) |

Table 13: Summary of notation used in the diffusion complexity analysis.

*Proof.* The forward diffusion consists of three operations:

**(i) Noise injection.** Each step adds Gaussian noise to a $d$-dimensional feature vector:

$$\mathcal{F}_t = \sqrt{\alpha_t}\mathcal{F}_{t-1} + \sqrt{1 - \alpha_t}\,\varepsilon_t, \qquad \varepsilon_t \sim \mathcal{N}(0, I_d). \tag{19}$$

This is a linear operation in $d$, hence $O(d)$ per step and $O(Td)$ overall.

**(ii) Discrepancy computation.** The diffusion discrepancy uses discrete finite differences:

$$\Delta s_t = s_t - s_{t-1}, \qquad \Psi = \left( \sum_{t=1}^{T} \gamma_t (\Delta s_t)^2 \right)^{1/2}. \tag{20}$$

Each similarity computation $\langle F_t, G_t \rangle$ requires $O(d)$ time, giving $O(Td)$ total.

**(iii) GMM-based partitioning.** Posterior updates and EM steps for a fixed $K = 3$-component GMM incur $O(Kd) = O(d)$ cost but do not depend on $T$. As $K$ is constant, the contribution is $O(1)$ in asymptotic notation.

Combining all terms yields $O(Td)$ time and $O(1)$ parameter complexity. □

### C.2 REVERSE DIFFUSION COMPLEXITY

**Proposition C.2** (Reverse diffusion complexity)**.** *Let the denoising network use a bottleneck structure with weights $W_1 \in \mathbb{R}^{d \times h}$ and $W_2 \in \mathbb{R}^{h \times d}$. If attention and projection layers operate in the $h$-*

*dimensional bottleneck space, the time complexity per batch is $O(Tdh)$, with parameter complexity $O(d^2)$ assuming shared weights across diffusion steps. If attention is computed in full dimension, the cost becomes $O(Td^2)$.*

*Proof.* The reverse diffusion at step $t$ computes

$$\hat{\mathcal{F}}_t = \text{LN}\left(\hat{\mathcal{F}}_{t-1} + W_{\downarrow}\,\sigma\left(W_{\uparrow}\,\hat{\mathcal{F}}_{t-1}\right) + \text{Attn}(Q_t, K_t, V_t)\right), \tag{21}$$

where $\sigma$ denotes a pointwise nonlinearity.

**(i) Bottleneck MLP.** Multiplications $W_{\uparrow}\hat{\mathcal{F}}_{t-1}$ and $W_{\downarrow}(\cdot)$ each cost $O(dh)$, giving $O(dh)$ per step.

**(ii) Cross-modal attention.** If attention is computed in the $h$-dimensional space, QKV projections and the attention score computation cost $O(dh)$ per step. If full-dimensional attention is used, the cost becomes $O(d^2)$.

**(iii) Consistency losses.** The intra-modal reconstruction loss costs $O(d)$ and the cross-modal contrastive loss costs $O(Bd)$, both lower-order terms.

Thus the per-step cost is $O(\max\{dh, d^2\})$, and over $T$ steps,

$$O(T\max\{dh, d^2\}). \tag{22}$$

With the bottleneck attention design of DiffNCL, this simplifies to $O(Tdh)$.

**Parameter complexity.** The weights $W_{\uparrow}$ and $W_{\downarrow}$ contribute $O(dh)$ parameters, while QKV projections contribute $O(d^2)$. Since $d^2 > dh$, the parameter complexity is $O(d^2)$ under step-wise weight sharing.

$\square$

## C.3 Overall Complexity of the Diffusion Module

**Corollary C.3** (Overall time and parameter complexity). *The overall time complexity of the diffusion module in DiffNCL is*

$$O(Tdh), \tag{23}$$

*since the reverse diffusion stage (with bottleneck dimensionality $h > 1$ and $h < d$) dominates the forward diffusion cost $O(Td)$. The parameter complexity is $O(d^2)$.*

*Proof.* From Propositions C.1 and C.2, the forward diffusion cost is $O(Td)$, while each reverse diffusion step requires $O(dh)$ computation. Because $h > 1$, we always have $dh > d$, and thus the reverse diffusion stage dominates the total runtime. Over $T$ diffusion steps, the resulting time complexity is $O(Tdh)$. The parameter complexity is governed by the $O(d^2)$ attention projection matrices, which dominate the $O(dh)$ parameters from the bottleneck MLP. $\square$

## C.4 Consistency with Empirical Observations

The theoretical scaling laws above are consistent with empirical timing results from B.1: (i) training cost grows approximately linearly in $T$, (ii) reducing the bottleneck ratio $h/d$ reduces runtime and parameter count.

# D The Use of Large Language Models (LLMs)

In this research, LLMs were used only as a general-purpose writing aid, without playing any role in core research ideation or technical processes. The use of LLMs was limited to polishing English academic expression, without altering any technical content. All LLM-assisted revisions were strictly verified by the author team to ensure accuracy, scientific rigor, and no misconduct. The author team takes full responsibility for the paper's content. LLMs are not contributors, ineligible for authorship, and not listed as authors.

