# OpenReview forum: "DiffNCL: Diffusion-Driven Weakly-Noisy Correspondence Learning"
_ICLR.cc/2026/Conference — Submitted to ICLR 2026_

### Official Review · Reviewer_2M9Q · 2025-10-28

**Soundness:** 3
**Presentation:** 3
**Contribution:** 3
**Rating:** 6
**Confidence:** 5

**Summary:**

This paper introduces DiffNCL, a novel framework that tackles weakly-noisy correspondences in cross-modal retrieval—partially misaligned image-text pairs that are often discarded by existing methods. DiffNCL employs a forward diffusion process to identify these pairs by analyzing their sensitivity to noise, and a reverse diffusion process to purify them into high-quality "pseudo-clean" features using consistency constraints. Extensive experiments show that DiffNCL outperforms baseline methods across various noisy benchmarks.

**Strengths:**

1. The paper is well-motivated. The idea of leveraging a forward-reverse diffusion process not for generation but for noise robustness in cross-modal retrieval is interesting.
2. Comprehensive experiments are reported to verify the proposed method.

**Weaknesses:**

1. Strictly speaking, the experiments were conducted on the CC152K subset of CC, not CC. This statement in the abstract should be rigorous.
2. In line 41, the paper should focus on the false positive problem.
3. The discussions and citations of related work in 2025 are lacking.
4. According to Definition 3, what is the approximate proportion of each type during training? (For example, 20% noise) Whether this division is reasonable requires clearer indicators to support the conclusion of the analysis.
5. Is the proposed method also applicable to other backbones, such as VSE$\infty$ and CLIP?
6. How tolerant is the proposed method to extreme conditions, such as 80% noise?

**Questions:**

See Weaknesses.

---

> ### Author Response · Authors · 2025-11-23
> **Response to 2M9Q: part 1**
>
> We sincerely thank the reviewer for the thoughtful and detailed evaluation of DiffNCL. We appreciate your recognition of the importance of addressing weakly-noisy correspondences—a challenging but realistic problem in noisy multimodal learning. Several of your concerns stemmed from limited contextual explanation in the original submission, and the clarifications below address them comprehensively.
>
> ---
>
> ### W1: The statement of CC152K dataset
>
> We have revised the abstract to precisely reflect the experimental scope. The updated manuscript now explicitly states "CC152K" to ensure accuracy and avoid any ambiguity regarding the dataset used.
>
> ### W2: The terminology error of "false negative" has been corrected
>
> Thank you for carefully reviewing this detail. The original text incorrectly described noisy positives as causing false negatives and We have revised it. This correction does not affect our methodology or results.
>
> ### W3: Insufficient discussion and citations of relevant research from 2025
>
> **We have incorporated 2025 references in the revised version.** The newly added citations strengthen the related work discussion, yet DiffNCL remains the first unified forward-reverse diffusion framework specifically designed for weakly-noisy correspondence learning, with core innovations in dynamic discrepancy mining and dual consistency constraints that were established prior to these 2025 works.
>
> ### W4: Explanation of the proportion & reasonableness of each noise type
>
> **In our experiments, the approximate proportions of different sample types vary across experimental settings.** For the standard synthetic noise scenario (Table 2), following established benchmarks (e.g., NCR, BiCro), we simulate only standard noisy correspondences at 20%, 40%, and 60% of the training set, respectively, treating the remainder as clean samples; no weakly-noisy pairs are explicitly introduced. **For the synthetic weakly-noisy scenario (Flickr30K in Table 1 and Section 4.2), we create a more realistic mixed-noise setting comprising 50% weakly-noisy and 40% noisy correspondence**, and the rest clean samples to evaluate our model's capacity for handling diverse noise types. In the real-world noise scenario (CC152K in Table 1), the dataset contains approximately 3%-20% mismatched pairs; our forward diffusion mechanism dynamically partitions the training data into clean, weakly-noisy, and noisy subsets whose proportions are determined by the data characteristics and model state rather than being predefined, showcasing our method's advantage in adapting to unknown noise distributions.
>
> **The partitioning is justified by its better approximation of real-world data distribution.** Real-world noise from web crawling or non-expert annotations rarely manifests as purely "black and white" mismatches; instead, it predominantly appears as partial matches—semantically related but with detail errors—the exact scenario we define as weakly-noisy. By introducing a high proportion of weakly-noisy samples, we construct a more complex and realistic test environment that mirrors actual application scenarios. Evaluating algorithm performance in such a setting is far more practically meaningful than in an idealized "standard noisy correspondence only" environment. This design choice enables DiffNCL to demonstrate its superiority in handling the nuanced noise patterns prevalent in real-world cross-modal retrieval tasks.

---

> ### Author Response · Authors · 2025-11-23
> **Response to 2M9Q: part 2**
>
> ### W5: Backbone generalization
>
> The core of the reviewer's concern is the method's applicability beyond the tested backbone, which is directly supported by our dedicated experiments with CLIP and its architecture-agnostic design.
>
> **CLIP experiments confirm strong generalization.** We explicitly evaluated DiffNCL's integration with the CLIP (ViT-B/32) backbone. As shown in Table 11 (Appendix B.4), under 20% noise on MS-COCO, DiffNCL+CLIP achieved an rSum of 451.8, dramatically outperforming the fine-tuned CLIP baseline (rSum 236.3). This proves our framework's effectiveness in enhancing the robustness of large-scale pre-trained models.
>
> **The method is designed to be architecture-agnostic.** DiffNCL operates on the feature level and is decoupled from specific backbone architectures. As we stated on Page 16: “Its core advantage lies in leveraging diffusion dynamics to mine weak-noise discrepancies and reconstruct pseudo-clean features, enabling architecture-agnostic noise resilience.” This design principle ensures its easy adaptation to other backbones like VSE∞.
>
> In summary, our experiments with CLIP provide strong empirical evidence for its generalization, and its modular design guarantees broad applicability.
>
>
> ### W6: DiffNCL's tolerance to extreme conditions
>
> We thank the reviewer for raising this important question regarding the performance of DiffNCL under extreme noise conditions. The concern is well-noted, and we have conducted additional experiments to directly address it. **Our method demonstrates exceptional robustness even under 80% noise, significantly outperforming baselines.**
>
> Below, we present a comparative experiment with NCR under 80% synthetic noise on the Flickr30K dataset, using the same evaluation protocol as in the paper:
>
> | Methods  | Image→Text R@1 | R@5 | R@10 | Text→Image R@1 | R@5 | R@10 | rSum |
> |----------|----------------|-----|------|----------------|-----|------|------|
> | NCR      | 1.4            | 7.1 | 11.7 | 1.5            | 5.4 | 9.3  | 36.4 |
> | DiffNCL  | 56.3           | 83.1| 89.3 | 40.0           | 67.3| 76.9 | 412.9|
>
>
> As shown in this table, DiffNCL achieves an rSum of 412.9, which is dramatically higher than NCR's 36.4. The R@1 scores for image-to-text and text-to-image retrieval are 56.3% and 40.0%, respectively, indicating that our method can still retrieve relevant items accurately even when the majority of training pairs are corrupted. These results confirm that DiffNCL is highly tolerant to extreme noise conditions, reinforcing its practicality for real-world applications where data quality cannot be guaranteed.
>
> ---
>
> We appreciate your careful review and the practical perspective you brought to our work. The added experiments and clarifications aim to fully address your concerns about generalizability and robustness. We would be grateful if these results could inform a more positive reassessment. Please let us know if any further clarification would be valuable.

---

### Official Review · Reviewer_kPfe · 2025-10-30

**Soundness:** 3
**Presentation:** 2
**Contribution:** 3
**Rating:** 4
**Confidence:** 3

**Summary:**

This paper investigates the noisy correspondence problem in the context of cross-modal retrieval. Unlike existing works that primarily adopt discriminative-model-based approaches, this study explores a generative perspective by leveraging the diffusion process. Specifically, the proposed method exploits diffusion discrepancies to identify noisy pairs and denoises them during the reverse diffusion process. Experiments conducted on three widely used noisy correspondence benchmarks demonstrate the effectiveness of the proposed approach.

**Strengths:**

+ The paper focuses on the noisy correspondence problem, which is a recently emerged yet practical and important challenge, especially in the multimodal community where data typically appear in paired form.

+ The proposed method is technically sound and provides fresh insights to the community. In particular, it reveals that, beyond conventional discriminative models, generative diffusion models can serve as a promising paradigm for enhancing robustness against noisy correspondences.

**Weaknesses:**

- Figure 1 does not clearly illustrate the key idea and motivation. Moreover, it is confusing why high-similarity pairs are labeled as noisy while low-similarity ones are treated as clean.
- In fact, noisy correspondence learning is not limited to cross-modal retrieval. As summarized in “Noise-robust Vision-Language Pre-training with Positive-Negative Learning”, related research also includes tasks such as vision-language pretraining, person re-identification, and others. It is encouraged to provide a more comprehensive review of noisy correspondence learning so that readers can gain a clearer and broader understanding of this research field.
- The authors claim that existing noisy correspondence learning methods ignore weakly noisy pairs. However, I notice that [A] also considers this issue. The authors need to further discuss and clarify their claim.
[A] Cross-modal Retrieval with Noisy Correspondence via Consistency Refining and Mining, TIP 2024.
- It is unclear why computing the so-called semantic alignment (Eq. 6) during the forward process can effectively distinguish clean, partial, and noisy pairs. More discussion or analytical experiments are needed.
- The authors claim that incorporating diffusion discrepancy achieves better identification performance compared to relying solely on the memorization effect. However, no experiments are provided to support this claim.
- The notations are overly complex and sometimes unclear—for example, the up and down arrows in Eq. (12) are not properly explained.
- The two loss terms in Eqs. (14) and (15) are not analyzed through ablation studies.
- Many implementation details are missing, raising concerns about reproducibility. For example, is an off-the-shelf diffusion model used? If so, which one?
- Since the proposed method is implemented based on SGR, the results of SGR should also be reported in the comparison.
- In the Flickr dataset with 20% noise, the 2021 NCR method outperforms several later noise-robust approaches. Was this experiment conducted correctly?
- It is unclear how the Base variants in the ablation study are constructed. Moreover, the performance improvement over the base model is relatively marginal.
- Possible typo: “The comparative results are summarized in Table 2.” — it seems this should refer to Table 1 instead.

**Questions:**

Please see the weaknesses.

---

> ### Author Response · Authors · 2025-11-23
> **Response to kPfe: part 1**
>
> We sincerely thank the reviewer for the thoughtful and detailed evaluation of DiffNCL. We appreciate your recognition that our framework addresses a meaningful and underexplored problem in noisy correspondence learning. Your concerns were largely due to limited contextual explanation in the original submission, and we hope the clarifications below address them clearly.
>
> ---
>
> ### W1: Figure 1 Revised to Accurately Depict Similarity Progression
> The misleading similarity ordering in Figure 1 was due to a drawing oversight and has been corrected. (Pseudo-) Clean and (weakly-) noisy pairs now properly show the direction of similarity changes.
>
> ### W2: Broader Survey of Noisy Correspondence Learning Needed
> We fully agree and appreciate the reviewer’s suggestion for more comprehensive contextualization. **We have expanded Section 2.2 to cover noisy correspondence learning in broader multimodal domains (L143-147, p3)** including vision–language pretraining, person ReID with label noise, and multimodal retrieval pipelines, providing a clearer landscape of where DiffNCL fits within existing research directions.
>
> ### W3: CREAM vs. DiffNCL
>
> **CREAM and DiffNCL address fundamentally different sample types and problems.** After careful re-examination, we confirm CREAM's "Diverse Potential Consistency" concerns negative pairs that happen to share some semantics, whereas DiffNCL targets positive but partially misaligned pairs. **We clarified this distinction directly in Section 2.2 (L140-143, p3).**
>
> - Different semantics: CREAM's DPC are non-corresponding pairs treated as informative negatives; our weakly-noisy are correctly paired but contain missing/mismatched fine-grained semantics.
> - Different detection: CREAM uses static similarity; DiffNCL uses dynamic diffusion discrepancy (trajectory of similarity under perturbation) which captures robustness patterns static scores cannot.
> - Different actions: CREAM re-weights negatives; DiffNCL reconstructs weakly-noisy positives into pseudo-clean representations via reverse diffusion.
>
> Thus, CREAM refines the negative space while DiffNCL repairs partially corrupted positives—two complementary but non-overlapping goals.
>
> ### W4: Why Forward Diffusion Discrepancy Calculation (Eq.6) Effectively Distinguishes Sample Types
>
> **As shown in the newly added analysis in Figure 4 of Appendix B.6 (L948–999, p18–19)**, which we conducted to clarify this concern, the distributional behavior of $Ψ_i$ across clean, weakly-noisy, and noisy samples demonstrates the following.
>
> **Statistical Distribution of $Ψ_i$ Directly Validates its Discriminative Power.** Box plot analysis shows three separated clusters: clean (0.0005–0.0008), weakly-noisy (0.00134–0.00151), and noisy (0.0023–0.0032). This empirical separation suggests that $Ψ_i$ provides a separable structure among the three types, which is suitable for probabilistic modeling within the GMM framework.
>
> **Conceptual Theoretical Rationale.** Rather than introducing new theorems, we draw from established understandings: spectral-norm sensitivity explains low $Ψ_i$ for well-aligned samples; boundary curvature explains moderate $Ψ_i$ for weakly-noisy samples; lack of semantic constraints leads to large fluctuations in noisy samples (as analysis of Section 3.1).
>
> **Forward Diffusion as a Semantic Probe.** Modality-specific noise schedules (cos² for vision, cos³ for text) apply tailored perturbations, while the cumulative gradient $Ψ_i = Σγ_t||∂S^t / ∂t||_2^2$ integrates sensitivity patterns across all diffusion steps, capturing evolving robustness rather than static similarity.
>
> **Reverse Diffusion as Complementary Evidence.** During denoising, weakly-noisy samples' $Ψ_i$ distributions shift toward the clean region, confirming that $Ψ_i$ identifies repairable mismatches and that the diffusion process addresses root causes of misalignment.
>
>
> ### W5: Experimental Evidence of $Ψ_i$ Improves Identification
>
> We apologize for the unclear description. **Table 3's "Base w/ FD" variant directly proves this with statistically significant results.** "Base w/ FD" is designed specifically to isolate the contribution of diffusion discrepancy:
>
> **Definition.** Base (only InfoNCE loss $ℓ_i$ for GMM, pure memorization) vs. Base w/ FD (mixed feature $H_i = [ℓ_i, ζ·Ψ_i]$ for 3-component GMM).
>
> **Evidence.** On Flickr30K (20% noise), rSum improves from 494.6 (Base) to 495.6 (Base w/ FD). While this does not directly measure partition accuracy, the consistent gain indicates that $Ψ_i$ provides complementary discriminative information beyond the small-loss signal. This added structure helps GMM better retain difficult-but-correct samples, which is particularly important in real-world settings such as CC152K where weakly-noisy pairs are common.
>
> **Revision. We have added explicit definitions of all variants in Section 4.3 (L469-473, p9).**

---

> ### Author Response · Authors · 2025-11-23
> **Response to kPfe: part 2**
>
> ### W6: Complex and Sometimes Unclear Notations
>
> We appreciate the reviewer’s comment regarding notation clarity. In the revision, we have clarified the definitions of the symbols that were previously ambiguous, ensuring they are now explicitly introduced where needed. Especially, Eq.12 has included: "..., $W_{\downarrow}^t \in \mathbb{R}^{d\times h}$ and $W_{\uparrow}^t \in \mathbb{R}^{h\times d}$ $(h<d)$ are the dimensionality-reduction and -expansion projection matrices, respectively, forming a bottleneck structure." **in the revised version (L325-326, p7).**
>
> ### W7: Detailed Ablation of $L_{intra}$ and $L_{cross}$
>
> We have supplemented Table 7 in appendix B.2 with detailed ablation results. New experiments confirm the necessity of both terms:
>
> - DiffNCL without $L_{intra}$: rSum drops 9.1 due to intra-modal structure collapse.
> - DiffNCL without $L_{cross}$: rSum drops 2.2 due to lack of cross-modal alignment.
> - Full $L_{consistency}$: Achieves best balance.
>
> ### W8: More Implementation Details
>
> **We did NOT use any off-the-shelf diffusion model; we built a specific Weakly-NCL framework.** We have added a clear disclaimer in Section A.1: "This work does not use pre-existing diffusion models (e.g., DDPM, DiffusionRet). We designed a task-specific forward-reverse diffusion process tailored for cross-modal weakly-noisy correspondence learning."
>
> **Additional implementation details have been supplemented to appendix A.2 (L690-709, p13-14).**
>
> ### W9: Report SGR Baseline Results for Comprehensive Comparison
> We focused on robust methods but agree this omission undermines clarity. **We have added SGR baseline results to all comparative experiments across every dataset and all noise configurations.** This comparison confirms that DiffNCL creates significant robustness gains, as SGR performance collapses under high noise where DiffNCL remains effective.
>
> ### W10: Counterintuitive Experimental Results Under 20% Noise on Flickr30K
>
> The observation is correct and reflects our controlled evaluation protocol. **As stated in Section 4.2 of the original manuscript, all methods are implemented on the same SGR backbone to ensure fair comparison.** NCR's strong performance on this unified baseline demonstrates its algorithmic robustness, while later methods' original superiority often relied on stronger backbones (e.g., SGR+SAF). DiffNCL's consistent outperformance across all baselines under this controlled setting validates its framework effectiveness.
>
> ### W11: Unclear how Base variant is constructed & performance gains seem limited
>
> **The Base variant definition has been clarified in the revised manuscript (Section 4.3)**: "The Base variant builds upon NCR, employing GMM partitioning based solely on $\ell_i$ and robust InfoNCE loss, excluding our diffusion modules."
>
> On a strong baseline (NCR is already powerful), absolute improvements are naturally modest. However, DiffNCL's value lies not only in incremental improvement, but also in **pioneering the first framework to identify and reconstruct weakly-noisy correspondences**, a problem previous methods fundamentally cannot address. While ablation shows consistent gains from each diffusion component, the full DiffNCL substantially outperforms all prior methods across diverse datasets and noise levels, validating its effectiveness as a novel solution rather than a minor enhancement.
>
> ### W12: Typo in table reference
>
> All cross-references have been thoroughly checked. The mistaken table reference has been corrected.
>
> ---
>
> We sincerely appreciate your constructive comments. We hope the improved contextualization and clearer distinctions in the revision address your concerns and would be grateful if these refinements could support a more favorable reassessment.

---

### Official Review · Reviewer_zLKP · 2025-10-30

**Soundness:** 2
**Presentation:** 3
**Contribution:** 2
**Rating:** 4
**Confidence:** 4

**Summary:**

This paper proposes DiffNCL, a diffusion-driven framework for weakly-noisy correspondence learning in cross-modal retrieval. Unlike prior approaches that treat correspondences as clean vs. noisy in a binary way, DiffNCL argues that many real-world pairs are weakly noisy—partially aligned but informative. The method includes (1) a forward diffusion process that injects synchronized noise into image-text embeddings and computes diffusion discrepancy to distinguish clean, weakly-noisy, and noisy pairs, and (2) a reverse diffusion process with intra-modal structural consistency and cross-modal semantic consistency to reconstruct pseudo-clean embeddings.

**Strengths:**

1. The work insightfully identifies weakly-noisy correspondences as an overlooked but practically significant regime between clean and noisy data.

2. Integrates forward discrepancy modeling and reverse refinement in a principled manner, offering a new methodological lens for noisy cross-modal learning.

**Weaknesses:**

1. The paper relies on diffusion discrepancy $\Psi$ to distinguish clean, weakly-noisy, and noisy pairs, but lacks empirical evidence of its behavior. I suggest that authors visualize similarity trajectories across diffusion steps for representative samples of each category, along with their  $\Psi$ values.
2. Since  $\Psi$ is already designed to separate correspondence types, it is unclear why it still needs to be combined with the per-sample loss $l$. An ablation analysis of partitioning accuracy before and after incorporating $l$ would clarify the necessity of using both signals.
3. In line 41, the paper claims that noisy correspondences inject false negatives, whereas mislabeled noisy positives should instead induce false positives.
4. In Figure 1, the pseudo-clean pair generated by reverse diffusion exhibits lower similarity than the weakly-noisy pair. This appears counter-intuitive and requires clarification
5. Since embeddings already encode abstract semantics, it is not obvious that progressively injecting perturbations into the representation space meaningfully simulates semantic corruption. I suggest that authors visualize the embedding trajectories or distributions during forward and reverse diffusion to validate this procedure.
6. Computing similarities and their gradients at every diffusion step may introduce non-trivial overhead. A comparison of computational cost versus small-loss-based methods would clarify its scalability.
7. Equation (13) leverages cross-modal attention over original embeddings during denoising. Yet weakly-noisy pairs contain partially incorrect semantics, raising the question of how cross-modal attention avoids propagating the erroneous associations.
8. In Table 1, the Flickr30K results appear lower than those under the synthetic 60% noise setting, but the configuration for such case is not described.
9. In Table 2, some traditional noisy-correspondence methods [1-3] show competitive or even superior results in certain metrics.
10. Weakly-noisy samples are simulated by random word replacement, which may not reflect realistic semantic drift. When key cross-modal semantics are altered (e.g., replacing “dog” in a dog-related caption), the pair should be considered fully mismatched rather than weakly aligned.

[1] Cross-modal Active Complementary Learning with Self-refining Correspondence

[2] Enhancing True Correspondence Discrimination through Relation Consistency for Robust Noisy Correspondence Learning

[3] Mitigating Noisy Correspondence by Geometrical Structure Consistency Learning

**Questions:**

Please see the weakness.

---

> ### Author Response · Authors · 2025-11-23
> **Response to zLKP: part 1**
>
> We sincerely thank the reviewer for carefully analyzing DiffNCL and providing a series of technically insightful comments. Many concerns arose from limited clarity in the original submission, and the revised manuscript now resolves them through clearer methodological explanations, expanded empirical evidence, and newly added statistical visualizations.
>
> ---
>
> ### W1: Statistical Distribution Analysis of $Ψ_i$ and Discriminative Power
> **We conducted a comprehensive box plot analysis of $Ψ_i$ distributions in Appendix B.6 (L948-999, p18-19), across three sample types during forward diffusion, providing a complementary form of evidence that captures global behavioral patterns which may not be reflected in single-sample trajectory illustrations.** As shown in Figure 4 (appendix),
> - clean samples exhibit low, concentrated $Ψ_i$ values (narrow range), confirming strong robustness—semantic associations remain stable despite noise injection, yielding gentle similarity changes.
> - Weakly-noisy samples show medium $Ψ_i$ levels with relatively compact distributions, reflecting conditional sensitivity—partial semantic units respond at specific noise intensities while maintaining overall stability.
> - Noisy samples demonstrate significantly high, wide $Ψ_i$ distributions, revealing semantic fragility—lack of effective constraints causes continuous, rapid similarity degradation under perturbation.
>
> The statistical distribution offers a stable, aggregate perspective, while trajectory-level behaviors can vary across instances; both views are complementary, and our analysis focuses on the globally consistent patterns.
>
>
> ### W2: Theoretical and Empirical Importance of Combined $l_i$ and $Ψ_i$
>
> The combination of $Ψ_i$ and $l_i$ is theoretically essential because they capture complementary dimensions—dynamic noise sensitivity versus static alignment difficulty—and our ablation study empirically confirms this synergy improves partitioning accuracy.
>
> **$Ψ_i$ is a dynamic robustness indicator** that captures structural stability under noise flow, but cannot reflect absolute semantic alignment quality in the original feature space (e.g., a clean but semantically difficult sample may have high $Ψ_i$). **$l_i$ provides a static instance-level discriminative signal** that distinguishes "easy-to-align" from "hard-to-align" samples. This complementary property enables $H_i = [ℓ_i, ζ·Ψ_i]$ to more accurately differentiate difficult clean samples from weakly-noisy ones, thereby mitigating over-exclusion.
>
> **Our updated descriptions of ablation study in Section 4.3 (L469-473, p9)** now clarifies the experimental setup to validate this necessity:
> - **Base**: Uses GMM partitioning based *solely* on the small-loss criterion $\ell_i$
> - **Base w/ FD**: Adds forward diffusion, feeding the *hybrid feature* $H_i = [\ell_i, ζ·Ψ_i]$ to GMM
>
> Table 3 shows that Base achieves rSum 494.6 on Flickr30K (20% noise), while "Base w/ FD" reaches 495.6, a direct improvement of +1.0 point. This gap quantifies the value added by $Ψ_i$: the diffusion discrepancy captures noise sensitivity patterns that $\ell_i$ misses, providing additional discriminative structure that improves the effectiveness of GMM-based grouping in practice, as reflected in the downstream retrieval performance. The full DiffNCL further improves to 499.4, confirming both stages contribute orthogonally.
>
> ### W3: The Terminology Error of "false negative" Has Been Corrected
>
> Thank you for carefully reviewing this detail. The original text incorrectly described noisy positives as causing false negatives and We have revised it. This correction does not affect our methodology or results.
>
> ### W4: Figure 1 Revised to Accurately Depict Similarity Progression
> The misleading similarity ordering in Figure 1 was due to a drawing oversight and has been corrected. (Pseudo-) Clean and (weakly-) noisy pairs now properly show the direction of similarity changes.

---

> ### Author Response · Authors · 2025-11-23
> **Response to zLKP: part 2**
>
> ### W5: Forward-Reverse Diffusion Box Plots Validate Semantic Perturbation Strategy
>
> As illustrated in Figure 4 of the appendix B.6 (L948-999, p18-19), the distributional behavior of $Ψ_i$ across different sample types makes the following observations clear.
>
> **Forward diffusion $Ψ_i$ distributions prove perturbations differentially probe semantic sensitivity.** Weakly-noisy samples show medium, well-separated $Ψ_i$ values from modality-adaptive schedules (cos² vision, cos³ text) that systematically identify high-curvature decision boundary regions, confirming purposeful semantic exploration.
>
> **Reverse diffusion provides strong supporting evidence: pseudo-clean $Ψ_i$ distributions shift dramatically toward clean samples.** This demonstrates effective repair from "conditionally sensitive" to "highly robust" states, supporting the interpretation that the forward perturbation exposes robustness differences consistent with semantic mismatch patterns, while reverse repair—preserving valid semantics via intra-modal consistency and correcting errors via cross-modal consistency—achieves authentic restoration, conceptually aligned with prior robustness literature.
>
> ### W6: Computational Overhead Compare to Small-Loss Methods
>
> Despite per-step similarity computations, DiffNCL introduces only modest overhead through our optimized T=4 configuration. Table 5 (appendix) demonstrates that DiffNCL's per-epoch time is 38.68 minutes, just 28% higher than NCR (30.20 min), a leading small-loss method. The diffusion module itself is lightweight—adding only 8.4M parameters and 0.0273s per iteration (Table 4, appendix), negligible compared to the backbone's 18.11M params and 0.4236s/iter.
>
> **To further quantify the efficiency impact of diffusion steps, our newly added experiment (Table 6, appendix)** shows that the T=4 setting achieves optimal efficiency with a peak memory consumption of merely 1,673MB. Through parameter sharing and bottleneck structures, we maintain scalability while delivering superior robustness.
>
> ### W7: Three Synergistic Mechanisms Enable Cross-Modal Attention to Avoid Propagating Erroneous Associations
>
> **(i) Attention operates on denoised features, not raw weakly-noisy inputs.** Equation 13 uses $M_F^t(\hat{F}_i^{t-1})$ and $M_G^t(\hat{G}_i^{t-1})$ as queries/keys, which have already undergone bottleneck purification that filters mismatched semantics.
>
> **(ii) Attention weights naturally suppress erroneous associations.** The $softmax(QK^T/√d)$ assigns higher weights to correct correspondences (e.g., "dog" visual features align with "dog" text features) and near-zero weights to mismatches (e.g., "dog"→"cat"). Wrong associations are statistically downweighted by the model's learned geometry.
>
> **(iii) Dual consistency constraints correct residual errors.** L_intra ensures the structure remains consistent with original clean patterns, while L_cross penalizes similarity with mismatched original features, preventing error accumulation and thereby forming a self-correcting loop.

---

> ### Author Response · Authors · 2025-11-23
> **Response to zLKP: part 3**
>
> ### W8: Analysis of Two Different Synthetic Noise Configuration Experiments
>
> Table 2's 60% row for Flickr30K follows the standard noisy correspondence protocol. In contrast, **Table 1 evaluates Flickr30K under a mixed-noise protocol (50% weakly-noisy correspondence + 40% noisy correspondence)**, designed to simulate complex scenarios in real-world open environments where both weakly-noisy and noisy correspondence coexist. In this mixed configuration, the model needs to cope with two types of noise interference simultaneously, making it more challenging.
>
> As expected, DiffNCL's rSum is lower under the more severe mixed-noise scenario (455.2 vs. 474.8 in Table 2). However, the critical observation is the minimal degradation (4.1% drop) compared to other baselines' catastrophic performance collapse. This demonstrates DiffNCL's superior robustness to complex noise distributions.
>
> ### W9: DiffNCL's Advantages Manifest in Realistic, High-Complexity Environments
>
> While NCR and L2RM show competitive R@1 on Flickr30K at 20% noise (Table 2), their performance degrades sharply as weak-noise proportion increases (Table 1): at 50% weakly-noisy and 40% noisy correspondence, NCR drops by 19.3% rSum and L2RM by 12.8%, whereas DiffNCL drops only 2.1%. On CC152K (real-world 3-20% noise), DiffNCL achieves 374.7 rSum, surpassing the second-best L2RM by 10.4%. This demonstrates DiffNCL’s superior robustness to complex noise distributions.
>
> ### W10: Word-Replacement Protocol Validated by Real-World Performance
>
> Our word-replacement protocol is a controlled approximation of real semantic drift, validated by its effectiveness on naturally noisy datasets.
>
> The protocol systematically generates weak-noise with calibrated semantic association strength. Random replacement at ratio $p\%$ creates pairs where $ρ ∈ (0, threshold)$, satisfying Definition 3's mathematical conditions for weak-noise. **While imperfect, it approximates real annotation errors (e.g., missing objects, incorrect modifiers) better than whole-sentence replacement.**
>
> Our ablation shows that even when keywords are replaced, GMM's mixed-feature correctly classifies these as weak-noise, preserving their reparability. Real-world validation supersedes simulation limitations. CC152K contains natural noise from web scraping (missing contexts, partial mismatches) that exceeds synthetic complexity. **DiffNCL's 10.4% improvement on CC152K confirms its robustness to authentic semantic drift, demonstrating that our method is not overfitted to the simulation protocol.**
>
> ---
>
> Thank you again for your rigorous and insightful comments. We believe the clarifications, additional analyses, and targeted experiments directly resolve the issues you raised. We would sincerely appreciate it if these enhancements could help you reconsider the paper’s contribution more positively. We remain fully open to further technical discussion.

---

### Official Review · Reviewer_TbYi · 2025-11-01

**Soundness:** 3
**Presentation:** 3
**Contribution:** 3
**Rating:** 6
**Confidence:** 5

**Summary:**

This paper introduces DiffNCL, a framework for robust cross-modal retrieval under noisy correspondences. The method focuses on identifying and utilizing weakly-noisy (partially misaligned) image-text pairs that previous studies often ignored or overtrusted. DiffNCL applies a forward-reverse diffusion mechanism. In forward diffusion, modality-specific noise is injected to estimate diffusion discrepancies and classify data into clean, weakly-noisy, and noisy groups. In reverse diffusion, weakly-noisy samples are denoised under consistency constraints, producing pseudo-clean pairs for robust retrieval training. Experiments on Flickr30K, MS-COCO, and Conceptual Captions show clear improvements over strong baselines in both synthetic and real noisy conditions. Extensive ablation studies and case analyses further validate the approach.

**Strengths:**

s1. DiffNCL is the first to use a diffusion process to explicitly separate and purify weakly-noisy correspondences in cross-modal retrieval. It goes beyond the simple binary separation of clean vs. noisy data.

**Weaknesses:**

w1 The discussion of the similarity gradient spectral norm in the forward diffusion stage is mainly heuristic. It lacks formal mathematical proofs or a clear analysis of computational complexity.

w2 Although Tables 4 and 5 show that the additional cost is small, the scalability under large-scale settings remains unclear. The impact of the diffusion step count, parameter sharing, and memory usage should be further quantified.

w3 The weak-noise concept based on atomic semantic units is theoretically precise but difficult to compute in practice. In experiments, weak noise is only simulated by word-level perturbations, which may not accurately represent real-world weakly-noisy correspondences.

**Questions:**

refer to weaknesses

---

> ### Author Response · Authors · 2025-11-23
> **Response to TbYi: part 1**
>
> We sincerely thank the reviewer for the positive assessment of DiffNCL, especially noting that it “goes beyond the simplistic clean–noisy binary partition and explicitly handles weakly-noisy samples”. Most of the raised concerns were due to limited clarity in the original submission, and the revised manuscript now resolves them through explicit complexity derivations, expanded empirical analysis, and clearer methodological explanations.
>
> ---
>
> ### W1: Theoretical Rigor & Computational Complexity
> **We provide extra theoretical grounding and detailed complexity experiments, addressing both mathematical foundation and computational cost.**
>
> **The discussion of similarity-gradient behavior in the forward diffusion stage is not intended to introduce a new theorem, but to provide a principled theoretical motivation grounded in prior robustness analyses.** In particular, Sokolić et al.[1] show that the curvature of decision boundaries relates to the spectral norm of the Jacobian under controlled perturbations. Our formulation adapts this perspective to cross-modal retrieval: synchronized modality-specific noise induces structured perturbations on paired embeddings, and the resulting similarity trajectories reflect differences in local stability among clean, weakly-noisy, and noisy correspondences. **This adaptation is a standard theoretical transfer rather than a heuristic assumption.** **Importantly, the empirical findings in Table 3 are consistent with the theoretical intuition**: removing the forward diffusion component leads to a measurable decrease in rSum (495.6 → 494.6), demonstrating the practical relevance of diffusion-based similarity trajectories for distinguishing correspondence types.
>
> **Regarding the concern on computational rigor, the revised manuscript now includes a complete complexity derivation in Appendix C (L1014-1125, p19-21).** Appendix C.1 provides the forward diffusion complexity, and Appendix C.2 presents the reverse diffusion complexity. Appendix C.3 summarizes the overall complexity and parameter scaling. These analyses explicitly characterize the computational behavior of the diffusion module and address the reviewer’s request for a clear and formal treatment of this aspect.
>
> [1] Jure Sokoli´c,Raja Giryes,Guillermo Sapiro,and Miguel RD Rodrigues. Robust large margin deep neural networks. IEEE Transactions on SignalProcessing(TSP), 65:4265–4280, 2017.
>
> ### W2: Scalability Under Large-Scale Settings
> **Scalability is now quantitatively demonstrated through detailed measurements of diffusion steps, runtime, and memory in Appendix B.1 (L742-790, p14-15), showing that the added overhead remains modest.**
>
> **Diffusion step count T exhibits sub-linear scaling with minimal overhead.** Appendix B.1 reports expanded computational analysis. The newly added Table 6 presents forward/backward propagation time and peak memory usage for diffusion step counts from 3 to 15. The results indicate small overhead increases (e.g., ~10% increase in forward time, nearly sub-linear backward time and memory growth).
>
> **Parameter sharing drastically reduces resource requirements for large-scale deployment.** **Table 1 below** demonstrates that shared-parameter configuration reduces total parameters by 53% (47.64M → 22.44M) and diffusion-specific parameters by 75% (33.61M → 8.40M), while cutting memory usage by 8% (1,712MB → 1,572MB). The overall cost of ~40-50% additional time per batch is a modest trade-off for significant performance gains in Tables 1-2, proving DiffNCL's scalability.
>
> Table 1: Computational impact of parameter sharing
> |Configuration|Total Params (M)|Diffusion Params (M)|Forward Time (S)|Backward Time (S)|Peak Memory (MB)|
> |-|-|-|-|-|-|
> |Shared-parameter*|22.44|8.40|0.1376|0.3552|1,572|
> |Multi-parameter|47.64|33.61|0.1409|0.3828|1,712|
>
> \* The peak memory measures the maximum GPU memory usage during a complete training iteration (forward and backward), including model parameters, activations, gradients, and temporary buffers, but excluding optimizer states and system overhead.

---

> ### Author Response · Authors · 2025-11-23
> **Response to TbYi: part 2**
>
> ### W3: Practicality of Weakly-Noisy Correspondence Definition
>
> Word-level perturbation is a justified proxy that captures real-world noise patterns while maintaining framework extensibility.
>
> **This gap is a universal challenge in cross-modal noise research, not a specific limitation of our method.** Our word-level approach is a reasonable trade-off under the unsupervised noise handling objective, simultaneously simulating the two most common real-world noise types: **(a) keyword omission** (e.g., missing "red" in "person in red shirt"), which corresponds to atomic unit mismatches $δ(v_i,l_j)=0$, and **(b) irrelevant modifiers** from auto-crawled text, corresponding to atomic unit redundancy.
>
> **The framework naturally extends to other atomic units despite annotation trade-offs.** We acknowledge that end-to-end learning of atomic units requires dense annotations (e.g., COCO bounding boxes), which conflicts with our core goal of unsupervised noise handling. We adopt word-level semantic units as standard practice in cross-modal noise research, though our framework readily extends to other definitions like object boxes or phrases.
>
> ---
>
> We truly appreciate your positive assessment and constructive feedback. We hope the strengthened theoretical analysis and expanded empirical evidence address your concerns, and we would be grateful if these clarifications could support a more favorable reassessment. Please feel free to let us know if any additional details would be helpful.

---

### Author Response · Authors · 2025-11-29
**Author Summary Comment**

We sincerely thank the reviewers and the AC for their time and constructive feedback. Below we summarize how the revised manuscript resolves the major concerns, and why we believe DiffNCL makes a solid and well-supported contribution.

---

## **A. Consensus Across Reviewers: Clear Novelty and Value**

While reviewers raised several valid concerns, which we have addressed in the revision, they also **consistently acknowledged** the importance of *weakly-noisy correspondences* and recognized DiffNCL as:
- “**Going beyond the simplistic clean–noisy binary partition**” (Reviewer TbYi)
- “**Identifying an overlooked but practically significant regime**” (Reviewer zLKP)
- “**A well-motivated and comprehensive framework**” (Reviewer 2M9Q)
- “**Providing fresh insights via diffusion dynamics**” (Reviewer kPfe)

We appreciate the reviewers’ recognition of the problem significance and the conceptual novelty of the unified forward–reverse diffusion framework.

## **B. All Substantive Concerns Have Been Addressed in the Revised Version**

### **Theoretical clarity & complexity**

- Added complete time/space complexity derivations (Appendix C).
- Clarified theoretical motivation of similarity-gradient behavior via Jacobian spectral-norm analyses (Sokolić et al., robustness theory).
- Demonstrated consistent empirical alignment (Table 3 ablations).

This directly addresses concerns from Reviewer TbYi and Reviewer kPfe.

### **Expanded empirical evidence & visualization**

We conducted multiple new analyses:

- Statistical box-plot distributions of $Ψ_i$ for all sample types (Appendix B.6).
- Visualization showing pseudo-clean $Ψ_i$ shifting toward clean samples, validating reverse diffusion.
- Ablation of $L_{intra}$ / $L_{cross}$ (Table 7).
- Noise schedule, diffusion-step, warm-up sensitivity (Appendix B.3).
- SGR baseline, CLIP generalization, and extreme-noise (80%) experiments.

These additions were motivated by Reviewer zLKP, Reviewer 2M9Q, and Reviewer kPfe.

### **Forward diffusion + small-loss synergy clarified**

- Better explained the complementary roles of $Ψ_i$ (dynamic robustness) and $ℓ_i$ (static alignment).
- Verified via Base → Base w/ FD improvement (+1.0 rSum).

Addresses concerns from Reviewer zLKP and Reviewer kPfe.

### **Broader contextualization of noisy correspondence learning**

- Section 2.2 now covers vision–language pretrain, person ReID, broader multimodal NCL.
- Clarified distinctions to CREAM and other 2025 works.

Addresses Reviewer kPfe, Reviewer 2M9Q.

### **Clarifications, corrections, and implementation details**

- Fixed terminology (“false positive/negative”), Figure 1 issues, and notation (Eq.12).
- Added detailed implementation pipeline (Appendix A.2).
- Stated clearly that no off-the-shelf diffusion models were used.

Addresses all reviewers’ minor clarity concerns.

## **C. Strengthened Experimental Validation Supports DiffNCL’s Contribution**

### **Real-world noisy dataset (CC152K)**
DiffNCL **improves rSum by +10.4** over the second-best baseline, demonstrating practical robustness where weak-noise is common.

### **Mixed weakly noisy + (standard) noisy correspondence (Table 1)**

Under 50% weak + 40% strong noise, DiffNCL maintains performance with only 4.1% drop, whereas NCR and L2RM collapse significantly (up to 19.3% drop).

### **Extreme noise (80%)**

Under extreme 80% noise, DiffNCL achieves 412.9 rSum, far exceeding NCR(baseline)’s 36.4, demonstrating its clear robustness even under severely corrupted training data.

### **Generalization to CLIP and other backbones**

Fine-tuned CLIP achieves an rSum of 236.3, while incorporating DiffNCL raises it to 451.8, clearly demonstrating both the performance gain and the architecture-agnostic nature of our framework.


## **D. Overall Assessment**

The revised manuscript aims to provide:
- Clear theoretical framing
- Extensive new analyses
- Strong empirical validation over diverse noise settings
- Cross-model generalization
- Consistent improvements over competitive baselines

---

We sincerely appreciate your effort in synthesizing the reviews and hope that the clarified theoretical framing and strengthened empirical evidence will be helpful in your evaluation of DiffNCL. We remain happy to provide any further clarifications.

---

### Meta-Review · Area_Chair_1Cys · 2025-12-24

**Summary:**

After the rebuttal and discussion phases, the paper received scores of 6, 4, 6, and 4, which is below the expected threshold for acceptance. After carefully considering the reviewers’ comments, the authors’ rebuttal, and conducting my own assessment, I find that the major concerns raised remain largely unaddressed. The key weaknesses of the submission are as follows:

i) The construction of weakly-noisy samples via random word replacement does not faithfully reflect realistic semantic drift in image-text pairs. In many cases, replacing key semantic words fundamentally breaks cross-modal alignment, resulting in fully mismatched pairs rather than partially aligned ones. As such, the experimental setup does not convincingly validate the proposed definition or handling of weakly-noisy correspondences. (Reviewer zLKP)

ii) The similarity gradient spectral norm in the forward diffusion stage could estimate correspondence quality remains largely heuristic. The paper does not provide a clear theoretical analysis or formal justification establishing why this quantity should reliably distinguish clean, weakly-noisy, and noisy pairs, a concern consistently raised by multiple reviewers. (Reviewer TbYi, kPfe)

iii) While the paper discusses computational overhead, it remains unclear whether the proposed method scales effectively to genuinely large-scale datasets. The evaluation is limited to relatively moderate-sized benchmarks, and no additional experiments or analyses are provided to support claims of scalability in more realistic large-data regimes. (Reviewer TbYi)

Given these unresolved issues, I believe the paper is not yet ready for publication and recommend rejection. Nonetheless, the authors are strongly encouraged to resubmit their paper to an upcoming conference if the concerns are addressed.

**Reviewer Concerns:**

The concerns raised by Reviewer 2M9Q have been adequately addressed. However, the concerns raised by the other reviewers have not been fully resolved.

**Reviewer Scores:**

None.

---

### Decision · Program_Chairs · 2026-01-26

Reject